

# Development of BFMCOUPLER (v1.0), the coupling scheme that links the MITgcm and BFM models for ocean biogeochemistry simulations

Gianpiero Cossarini[1], Stefano Querin[1], Cosimo Solidoro[1], Gianmaria Sannino[2], Paolo Lazzari[1], Valeria
Di Biagio[1], Giorgio Bolzon[1]

[1]Department of Oceanography, Istituto Nazionale di Oceanografia e di Geofisica Sperimentale - OGS, Sgonico (TS), 34010,
Italy
[2]UTMEA-CLIM Laboratory, ENEA, Rome, 00123 Italy

*Correspondence to*: Gianpiero Cossarini (gcossarini@inogs.it)

**Abstract.** In this paper, we present a coupling scheme between the Massachusetts Institute of Technology general
circulation model (MITgcm) and the Biogeochemical Flux Model (BFM). The MITgcm and BFM are widely used models
for geophysical fluid dynamics and for ocean biogeochemistry, respectively, and they benefit from the support of active
developers and user communities. The MITgcm is a state-of-the-art general circulation model for simulating the ocean and
the atmosphere. This model is fully three dimensional (including the non-hydrostatic term of momentum equations) and
includes a finite-volume discretization and a number of additional features enabling simulations from global ($O(10^7)$m) to
local scales ($O(100)$m). The BFM is a complex biogeochemical model that simulates the cycling of a number of constituents
and nutrients within marine ecosystems. The coupler presented in this paper links the two models through an efficient
scheme that manages communication and memory sharing between the models. We also test specific model options to
balance the numerical accuracy against the computational performance. The coupling scheme allows us to solve several
processes that are not considered by each of the models alone, including light attenuation parameterizations along the water
column, phytoplankton and detritus sinking, external inputs, and surface and bottom fluxes. Moreover, this new coupled
hydrodynamic-biogeochemical model has been configured and tested against an idealized problem (a cyclonic gyre in a mid-
latitude closed basin) and a realistic case study (central part of the Mediterranean Sea in 2006-2012). The numerical results
are consistent with the expected theoretical and observed behaviour of both the idealized system and the Mediterranean
domain, thus demonstrating the applicability of the new coupled model to a wide range of ocean biogeochemistry problems.

## 1 Introduction

Coupling different models that have been specifically developed to study only limited aspects of the Earth's systems is
becoming increasingly common due to the need to simulate different environmental components – and their interactions –
simultaneously (Heavens et al., 2013). In marine ecosystems, physical processes influence the biogeochemistry by
modifying the spatial distribution of dissolved and suspended compounds via transport and mixing and by defining
temperature, irradiance and sea ice fields. Similarly, biogeochemical feedbacks on physical processes include the role of
particles in changing the transparency of water (hence its absorption of solar radiation) as well as the viscosity and density.
Therefore, coupled hydrodynamic-biogeochemical models are widely used to investigate and predict the physical,
biogeochemical and ecological properties of marine ecosystems across a wide range of scales and provide useful tools that
support environmental management and policies.

The numerical implementation of a coupling framework between three-dimensional hydrodynamic models and
biogeochemical models is not a trivial task (Bruggeman and Bolding, 2014) because every model focuses on processes that
occur on different temporal and spatial scales and uses different numerical parameterizations and schemes. Additionally,



these models might be coded in different languages or follow different coding 'philosophies' with respect to memory allocation, computational schemes and code workflow. Furthermore, hydrodynamic and biogeochemical models are often developed by different and highly specialized scientific groups, whereas coupling requires interdisciplinary expertise.

Because of the increasing availability of computational resources, hydrodynamic and biogeochemical models have
experienced significant improvements in recent decades in terms of the number of processes that are considered, the temporal and spatial resolution of the simulations, and the flexibility and modularity of the applications. In addition, the complexity of these codes has increased substantially, and model development is a cooperative and multidisciplinary task rather than an individual effort. A large number of generic, open-source models are utilized by the scientific community, and they can be customized to match the users' specific applications. A non-exhaustive list of the main state-of-the-art,
hydrodynamic community models includes the MITgcm (Adcroft et al., 2016), GOTM (Burchard et al., 2006), ROMS (Haidvogel et al., 2000) and NEMO (Madec, 2014), whereas examples of community biogeochemical models include the BFM (Vichi et al., 2015), ERSEM (Butenshon et al., 2015), PISCES (Aumont et al., 2015) and ERGOM (Neumann, 2000). Hydrodynamic and biogeochemical models can be coupled by merging their codes into a single larger new code, in which the original parts are intertwined. In this case, biological models are generally inserted into the workflow of the existing
hydrodynamic model (Burchard et al., 2006, Follows et al., 2006) because hydrodynamic codes are far more complex than biogeochemical codes. Alternatively, a modular approach is preferable because it facilitates upgrades for future releases of the original codes. In fact, each component preserves its own peculiarities, the coupling is performed only on localized portions of the code, and there are clear input/output directives. The separation of the two coupled components facilitates the maintenance of each code within its development community, avoids possible large efforts in solving the language
differences between models and eliminates the need to keep models up to date with respect to the parent model. As an example, Bruggeman and Bolding (2014) proposed a set of programming interfaces (FABM) that allows communication between different hydrodynamic and biogeochemical models.

In this paper, we present a coupling scheme between the MITgcm hydrodynamic model and the BFM biogeochemical model for ocean biogeochemical simulations. The MITgcm has already been coupled to low- (Parekh et al., 2005; Follows et al.,
2006) or intermediate-complexity (Hauck et al., 2013, Cossarini et al., 2015a) biogeochemical models for a few specific applications and to a specific high-complexity model (Dutkiewicz et al., 2009) to explore the theoretical aspects of intraspecific competition in plankton communities. However, this model has not yet been coupled to state-of-the-art multi-nutrient and multi-plankton models, which are the de facto standard in community biogeochemical modelling. The BFM has already been coupled to POM (Polimene et al., 2006), NEMO (Vichi and Masina, 2009; Epicocco et al., 2016) and OGSTM,
an upgraded version of OPA (Lazzari, et al 2012), although it has never been coupled to the MITgcm.

We develop a modular coupling scheme that maintains the portability and complexity of these two state-of-the-art models and jointly preserves the sustainability of the programming effort to handle future evolution in the codes. Coupling environments/tools were not used because we want to preserve and exploit the good computational performance of the MITgcm (Marshall et al., 1997). We demonstrate that the new coupled model provides reliable results when simulating
different marine ecosystems by correctly reproducing the interplay between physical, chemical and biological processes and components. The coupled model also runs with good computational performance and preserves the numerical accuracy of the solution. Therefore, the MITgcm-BFM model represents a promising tool for investigating marine biogeochemistry at different spatial and temporal scales.

This paper is organized as follows. After a brief presentation of the two models (section 2), we focus on the technical aspects
of the coupling algorithm. In the following section (section 3), we describe the testing of the new coupled hydrodynamic-biogeochemical model against the idealized case of a cyclonic circulation in a closed basin and against a real case study in



the central Mediterranean Sea. The paper closes with a discussion of the key issues of the coupling and future perspectives. A manual of the new code package is detailed in the Appendix.

## 2 Formulation of the hydrodynamic-biogeochemical coupling

A coupled hydrodynamic-biogeochemical model is composed of three main elements: a hydrodynamic sub-model, which

solves the governing equations for oceanic flows; a tracer transport sub-model, which solves for the transport (advection and diffusion) of biogeochemical variables (commonly called tracers); and a biogeochemical sub-model, which solves the relationships (i.e., biogeochemical reactions) among the biogeochemical variables.

Following the very common practice in which the biological feedback on transport is negligible, one can assume that changes in biogeochemical properties do not affect the water velocity, density or other physical properties; therefore,

modifying the standard equations that underpin hydrodynamic models is unnecessary. We adopted such an assumption for this numerical coupling framework; however, this coupler was developed, in principle, to also handle biological feedbacks on hydrodynamics.

The coupled model solves the set of partial differential equations specified below:

$$\frac{D\boldsymbol{v_h}}{Dt} + f\boldsymbol{k} \times \boldsymbol{v_h} + \frac{1}{\rho_c}\boldsymbol{\nabla}_h p' = \boldsymbol{F_h} \tag{1}$$

$$\epsilon_{nh}\frac{Dw}{Dt} + \frac{g\rho'}{\rho_c} + \frac{1}{\rho_c}\frac{\partial p'}{\partial z} = \epsilon_{nh}F_w \tag{2}$$

$$\boldsymbol{\nabla}_h \cdot \boldsymbol{v_h} + \frac{\partial w}{\partial z} = 0 \tag{3}$$

$$\rho' = \rho(\theta, S, p(z)) - \rho_c \tag{4}$$

$$\frac{D\theta}{Dt} = Q_\theta \tag{5}$$

$$\frac{DS}{Dt} = Q_S \tag{6}$$

$$\frac{D\boldsymbol{C}}{Dt} = \boldsymbol{Q_C} + \boldsymbol{R} \tag{7}$$

$$I = I(Q_{sw}, \boldsymbol{C}) \tag{8}$$

$$\frac{D}{Dt} = \frac{\partial}{\partial t} + \boldsymbol{v_h} \cdot \nabla \tag{9}$$

Momentum conservation equations, Eq. (1-2), continuity and density equations, Eq. (3-4), and active-tracers equations (for potential temperature $\theta$ and salinity $S$), Eq. (5-6), are formulated according to the semi-compressible Boussinesq approximation. In the equations, $\boldsymbol{v_h} = (u, v, 0)$ is the horizontal velocity, $w$ is the vertical velocity, $f$ is the Coriolis parameter, $\rho_c$ is a constant reference density, and $p'$ is the pressure term. The RHS terms in Eq. (1-2) and Eq. (5-6) correspond to the forcing and dissipation terms, including the diffusion, which acts on the momentum ($\boldsymbol{F}$ in Eq. (1-2)) and on

the temperature and salinity ($Q_\theta$ and $Q_S$ in Eq. 5-6). Similarly, Equation (7), which stands for a system of partial differential equations, encompasses the forcing and dissipation terms for biogeochemical tracers, $\boldsymbol{Q_C}$, and the biogeochemical reactions that occur in the sea, $\boldsymbol{R}$.

Eq. (8) is an equation of state that calculates the modulation of irradiance $I$ with depth starting from short-wave surface radiation fields ($Q_{sw}$). The total derivative accounts for the partial derivative in time and the advection term, which is related

to the flow field, Eq. (9).

By adopting a more explicit formulation and commonly used assumptions based on scale analysis (see Crise et al., 1999), Eq. (7) can be rewritten as follows:



$$\frac{\partial C}{\partial t} = -\boldsymbol{U} \cdot \nabla(\boldsymbol{C}) + \nabla_H\big(K_h \, \nabla_H(\boldsymbol{C})\big) + \frac{\partial}{\partial z}\Big(K_v \frac{\partial C}{\partial z}\Big) + w_{bio} \frac{\partial C}{\partial z} + \boldsymbol{R_{BFM}}(\theta, S, \rho, I, \boldsymbol{C}) \qquad (10)$$

The first three terms on the RHS of Eq. (10) represent the advection (first term) and diffusion (second -horizontal- and third - vertical- term) of biogeochemical tracers. The remaining terms describe the sinking processes that affect biological particles

(fourth term) and biogeochemical reactions (fifth term).

Within a coupled model, Eq. (1-6) are solved by the hydrodynamic sub-model, whereas Eq. (10) is solved partly by the transport sub-model, which is usually embedded into the hydrodynamic code, and partly by the biogeochemical sub-model. The other components, such as Eq. (8) and the sinking terms, can be handled by both the hydrodynamic and biogeochemical models.

A coupler is defined as the interface that transfers the hydrodynamic information from Eq. (1-6) to Eq. (10) and controls the communication between the different terms of Eq. (10). In this study, the sub-models coupled are the MITgcm (managing both hydrodynamics and transport) and BFM (for the biogeochemistry) models, which are described in Sect. 2.1 and 2.2. The algorithm used to construct the fully coupled system is detailed in Sect. 2.3.

**2.1 MITgcm**

The MITgcm (Massachusetts Institute of Technology general circulation model; Marshall et al., 1997) is a three-dimensional, finite-volume, general circulation model used by a wide community of researchers. The open source code of the MITgcm can be customized to create different simulation set-ups by modifying its packages and parameters accordingly (Adcroft et al., 2016). This model has already been successfully applied to a wide range of case studies for the world's ocean at different spatial and temporal scales. The MITgcm can solve fully non-hydrostatic and hydrostatic equations and can

handle different free surface formulations. Subgrid-scale turbulence in both the horizontal and vertical directions can be parameterized by using different types of closure schemes with either constant or variable coefficients (e.g., Gent-McWilliams, Redi, Leith, Smagorinsky, KPP and GGL90). MITgcm code is composed by several packages, and depending on the selected experiment, the compiled packages can be enabled or disabled during the runtime by specifying the flag use*PACKAGENAME*=.TRUE./.FALSE. in the data.pkg input namelist. The MITgcm's implementation in this paper was

based on the release 1 - checkpoint 65 k (April 2015) version of the code. Among the different available customization options, we adopted the fully implicit barotropic time stepping for the free surface, which is unconditionally stable. The vertical diffusion and viscosity terms in the horizontal momentum equations were treated implicitly in time and were solved by using a backward method. The terms that were evaluated explicitly were time discretized by using the third-order Adams–Bashforth method for the momentum equations and a forward-in-time method for the transport equations.

A native transport sub-model for passive tracers (Passive TRACERS -PTRACERS- package; according to the MITgcm's jargon, a passive tracer is a generic tracer that has no influence on the hydrodynamics - e.g., by changing the density and/or viscosity -) is included in the MITgcm code. This sub-model solves the first three terms on the RHS of Eq. (10) (transport of a generic passive tracer). This transport is calculated by adopting a direct space–time discretization method for the advection–diffusion part of the tracer equations and a nonlinear, third-order advection scheme with a Sweby flux limiter

(Sweby, 1984) to avoid spurious oscillations in the model output fields. When employing the direct space-time method and the flux-limited schemes, forward-in-time time stepping is adopted rather than Adams-Bashforth.

Because of the different length scales, horizontal and vertical turbulent processes are treated separately and are solved by adopting a selected subset of several available parameterizations: in this study, we chose a mixed Leith-Smagorinsky scheme for the horizontal processes (second term on the RHS of Eq. (10)) and the K-profile parameterization (KPP, Large et al.,

1994) for the vertical processes (third term on the RHS of Eq. (10)).





The packages that were enabled during compilation (#define ALLOW_*PACKAGENAME*) were the standard geophysical fluid dynamics packages of the MITgcm ("gfd": MOM_COMMON MOM_FLUXFORM MOM_VECINV GENERIC_ADVDIFF DEBUG MDSIO RW MONITOR), the oceanic packages ("oceanic": GMREDI and KPP), and our specific selections (TIMEAVE, CAL, EXF, OBCS, FLT, DIAGNOSTICS, PTRACERS and GCHEM), including the

coupling and long time-stepping packages (BFMCOUPLER and LONGSTEP), which are the core of this peculiar implementation.

This code was compiled onto a Linux cluster that was equipped with Intel Xeon Ivy Bridge processors by using both the native GNU compiler (gfortran with openmpi libraries) and the Intel compiler (ifort: Intel Composer XE 2013 SP1) and by adopting the optimization levels -O3 and -O2, respectively. Overall, the model performance increased by approximately

10% when using the Intel compiler. The results in this paper were obtained using the Intel compiler with the optimization set to -O2. Further details on the customary model installation are available on the MITgcm's online documentation (Adcroft et al., 2016 and http://mitgcm.org).

## 2.2 BFM

The Biogeochemical Flux Model (BFM) is a numerical model that was designed to describe the dynamics of the major

biogeochemical processes that occur in marine ecosystems. The standard configuration of the BFM solves the cycles of carbon, phosphorus, nitrogen, silica, and oxygen in the water-dissolved phase and in the plankton, detritus, and benthic compartments. Plankton dynamics are parameterized by considering a number of plankton functional groups, each representing a class of taxa. The BFM's plankton functional groups are subdivided into producers (phytoplankton), consumers (zooplankton), and decomposers (bacteria). These broad functional classifications are further partitioned into

functional subgroups to create a planktonic food web (e.g., diatoms, picophytoplankton, microzooplankton, etc.). The structure of the plankton functional types is modular and can be adapted to specific needs. In fact, the BFM's code is organized into several modules devoted to several plankton function types: PhytoDynamics (for the phytoplankton functional types), MesoZooDynamics and MicroZooDynamics (for the zooplankton functional types), and PelBacDynamics (for bacteria). The two modules OxygenReaerationDynamics and PelChemDynamics for the oxygen and carbonate system

dynamics, respectively, complete the pelagic system (subroutine PelagicSystemDynamics). The interface routine EcologyDynamics manages the memory allocation of the biogeochemical state variables and derivatives and the external information that is required to calculate the biological equations: temperature, salinity, presence of ice, wind, position of the cell with respect to the surface or bottom, and atmospheric $CO_2$ partial pressure. The code and a full description of the model equations and parameterizations are freely available at http://bfm-community.eu.

For this application, we adopted the configuration used in Lazzari et al. (2012), which is easily configured from the official release options (Vichi et al., 2015) and uses a zero-dimensional data structure for the biogeochemical state variables. The present BFM includes four components (C, N, P, and Si); four phytoplankton groups; four zooplankton groups; one group each of bacteria, detritus, labile and semilabile organic matter; and additional variables, such as dissolved oxygen and alkalinity (Fig. 1). In addition, chlorophyll is solved as a prognostic variable according to the formulation of Geider et al.

(1997), and the carbonate system is solved by using the OCMIP formulation (Melaku Canu et al., 2015, Cossarini et al., 2015b).

## 2.3 The coupler

In this coupling scheme, we adopted a modular approach by considering the high complexity of the two models that were employed. The size of the codes according to the SLOCCount tool (Wheeler, 2015) is approximately 400,000 code lines for

the MITgcm and approximately 20,000 for the BFM. The coupler is a package that handles the interface (input/output



directives) between the host code (MITgcm) and the BFM to solve Eq. (7-8) and to efficiently manage the matrices that contain the variables and tendencies shared by the two models and the flow of information among the different sub-model components.

The MITgcm-BFM coupling (Fig. 2) was achieved by upgrading the MITgcm package GCHEM (GeoCHEMistry), which
handles the evolution of tracers, and by developing an additional package, BFMCOUPLER, which was specifically designed as the interface with the BFM model. The BFM is called by the MITgcm as an external library; therefore, the BFM was compiled separately using the same compiler used for the MITgcm (additional details on the compilation options and instructions are provided in Appendix A).

The BFMCOUPLER package (grey shaded area in Fig. 2) manages the initialization and memory usage of the BFM. This
package also calls the BFM core routines and solves several processes that are not included in either model. The interfaces among the different components of the coupled model were designed so that the tracer transport sub-model (MITgcm PTRACER package) uses the $u$, $v$ and $w$ components of the velocity and the horizontal and vertical diffusivities ($K_h$ and $K_v$) from the hydrodynamic sub-model to integrate the transport equation. Furthermore, the transport sub-model must consider the boundary conditions along the open boundaries of the model domain (OBC)$_C$ and the surface fluxes, such as the mass
transport associated with the evaporation minus the precipitation minus the runoff term (E-P-R)$_C$.

As an interface, the BFMCOUPLER manages the one-way transfer of information that is required by the BFM solver from both the hydrodynamic and transport sub-models of the MITgcm. The values of the tracers are derived from the transport sub-model. Moreover, the hydrodynamic sub-model supplies the temperature, salinity and photosynthetic active radiation (PAR) values as well as additional forcing parameters (presence of ice, wind speed and air partial pressure of $CO_2$) and
information, such as the position of the specific grid cell within the water column (surface, intermediate or bottom), which activates specific processes (e.g., surface air-sea gas transfer or bottom sediment fluxes). Then, the BFM calculates the biological partial derivative of Eq. (10) (fourth term), and the BFMCOUPLER returns this term to the time integration package, which integrates the transport and biogeochemical derivative terms to solve Eq. (10). Certain information used by the BFM, such as the PAR and wind values, can be calculated directly from the internal variables of the hydrodynamic sub-
model (such as short wave radiation ($Qsw$) or other atmospheric fields) or managed from external sources.

### 2.3.1 Source splitting, processor splitting and longstep options

We considered several coupling strategies according to the MITgcm's code structure (Fig. 3). Within each time step of the model integration, which is coded in the routine FORWARD_STEP, the MITgcm solves the hydrodynamic equations (Eq. 1-6) through several routines: DO_ATMOSPHERIC_PHYS, DO_OCEAN_PHYS, DYNAMICS, and TEMP_INTEGRATE
and SALT_INTEGRATE; further adjustments for T and S (e.g., filters) are applied in TRACERS_CORRECTION_STEP (Adcroft et al., 2016).

Different options can be used to solve the evolution of tracers (Eq. 7), which can be controlled by the pre-compilation option "*gchem_separate_forcing*". When this option is false (*#ifndef* in Fig. 3), a source splitting scheme is adopted; therefore, the transport (dC$_{trsp}$) and biogeochemical (dC$_{bio}$) tendencies are calculated by using the same (current) values of the physical and
biogeochemical variables:

$$C_{n+1} = C_n + \left( \frac{\partial C_{\text{trsp}}(C_n, \boldsymbol{U}_n)}{\partial t} + \frac{\partial C_{\text{bio}}(C_n, T_n, S_n, f)}{\partial t} \right) * \Delta t \qquad (11)$$

where $\boldsymbol{U}$, $T$, $S$ and $f$ are the hydrodynamic variables and the additional forcings and $\boldsymbol{\Delta}$t is the time discretization, which is the same adopted by the hydrodynamic sub-model.





The biogeochemical tendency, which is solved by calling the BFM through the routine BFMCOUPLER_CALC_TENDENCY, is temporarily stored in the gchemTendency matrix, which is then summed to the overall tracer tendency, gTracers, in the routine PTRACER_APPLY_FORCING, along with the tendency term from the evaporation minus the precipitation effect (surfPtracers). The transport terms of the tracers (which update the gTracers

matrix) are subsequently calculated within the PTRACER_INTEGRATE routine by several routines (GAD_ADVECTION, GAD_CALC_RHS, IMPLDIFF and others) according to the options and numerical schemes selected in the specific MITgcm simulation setup. The TIMESTEP_TRACER routine calculates the integration of Eq. (10) by providing a new state for the tracers. However, when the MITgcm setup is prescribed with an implicit vertical diffusion scheme, an update of the state of tracers is solved within the IMPLDIFF routine according to the specific parameterization of the vertical diffusion

(e.g., KPP, GGL90). Finally, if open boundary conditions are prescribed in the MITgcm setup, the OBCS_APPLY_PTRACER routine applies the updated values of the tracers at the boundaries. The calculation of the derivative of the transport processes ($dC_{trsp}$) involves several MITgcm packages (GENERIC_ADVDIFF, PTRACERS, GCHEM, OBCS, KPP, and EXF) and options (choice of the advection scheme, viscosity and diffusivity coefficients, parameterization of surface dilution/concentration of tracers from evaporation, precipitation, and runoff), which are

exhaustively described in the MITgcm documentation (Adcroft et al., 2016).

For the second coupling option, a process splitting scheme is selected when "*gchem_separate_forcing*" is true *(#ifdef)*. In this case, the biogeochemical tendency ($dC_{bio}$) is calculated after the state of the tracers has been updated by the transport equation terms. An intermediated value of the tracers, $\widetilde{C_{n+1}}$, is passed to BFMCOUPLER_CALC_TENDENCY along with the values of the updated hydrodynamic variables (eq. 12).

$$\begin{cases} \widetilde{C_{n+1}} = C_n + \dfrac{\partial C_{trsp}(C_n, \mathbf{U}_n)}{\partial t} * \Delta t \\ C_{n+1} = \widetilde{C_{n+1}} + \dfrac{\partial C_{bio}(\widetilde{C_{n+1}}, T_{n+1}, S_{n+1}, f)}{\partial t} * \Delta t \end{cases} \tag{12}$$

This option allows for the development of an integration scheme with different time steps for the hydrodynamic and transport parts on one side and for the biological processes on the other.

A third option is a process splitting algorithm that involves the MITgcm package LONGSTEP (Adcroft et al., 2016) to adopt different time steps for the hydrodynamic and transport-biogeochemical components and to increase the computational performance of the entire code. In particular, the tracer time step is set as a multiple of the main (hydrodynamic) time step, $\Delta t_{trc} = LSn * \Delta t$, whereas the terms *U*, *T*, *S* and *f* in Eq. (11) are replaced by suitable averages of the physical variables. The calculation of the averages is controlled by the parameter *LS_when_to_sample*, which defines the position within the code

workflow in which the hydrodynamic variables are sampled (*longstep_average*, Fig. 3) and biogeochemical tracer tendencies are calculated (LONGSTEP_THERMODYNAMICS, Fig. 3*)*. To activate this option, the LONGSTEP package code must be modified properly: the LONGSTEP_THERMODYNAMICS routine must be modified by adding a call to the modified GCHEM_CALC_TENDENCY routine.

This third method is preferred over the previous one as a possible method of decoupling the numerical biogeochemistry

solution from the hydrodynamic solution. We tested the model to verify the trade-off between the increase in computational performance and the loss of accuracy in the model results as a function of the extension of the time step for the tracer equations (LSn, see section 3.1.3).

### 2.3.2 BFMcoupler processes

The core of the present coupling scheme is the new routine BFMCOUPLER_CALC_TENDENCY, which is called by

GCHEM_CALC_TENDENCY or by GCHEM_FORCING_STEP. The approach adopted in this coupling scheme is to loop





in space and to call the BFM as a subroutine to calculate the derivative terms of each biogeochemical tracer for each computational grid point (dC$_{bio}$ in Fig. 2). The derivatives of the chemical and biological processes are calculated by the BFM model via an Euler forward scheme through the BFM0D_ECOLOGY_DYNAMICS routine (a BFM routine) and stored in the 4D MITgcm matrix gchemTendency. Additionally, the contributions of other processes, which are not

explicitly coded in the BFM, are solved within BFMCOUPLER_CALC_TENDENCY, namely, the light penetration formulation, the sinking of phytoplankton and detritus, and the exchange processes in the surface and bottom layers of the water column.

In particular, the BFMCOUPLER package calculates the vertical profile of PAR along the water column, starting from the surface PAR, which is read from an external file or by using the shortwave radiation field (**Q$_s$**), which is converted into PAR

by a standard bulk formula, if the native MITgcm atmospheric forcing package EXF is active (Britton and Dodd, 1976):

$$PAR_s = Q_s * conv * pfrac \qquad (13)$$

where PAR$_s$ is the PAR at the sea surface, conv is a conversion factor of 4.6 µEin/m$^2$/s (W/m$^2$)$^{-1}$, and pfrac is the fraction of

the radiation in the visible band, which equals 0.4. The calculation of PAR along the water column, Eq. (14), is performed in the cell centre according to the Lambert-Beer formulation for the light exponential decay with depth and the shading of detritus and phytoplankton:

$$PARz = PARs \cdot e^{-\int_0^z \left(K_{ext} + \sum_j Pl_j \cdot K_{Pj} + R_C^{(3)} \cdot K_R\right) dz} \qquad (14)$$

where Pl$_j$ is the chlorophyll concentration of the j$^{th}$ phytoplankton functional type (PFTs), $R_C^{(3)}$ is the carbon concentration of the detritus or optically active organic matter, and Kp$_j$ and K$_R$ are the corresponding extinction factors. K$_{ext}$ represents a background value set constant and is equal to 0.035 m$^{-1}$ (considering pure water), or it can be read from an external file. In the latter case, the external file contains maps of the background extinction factor, which can be built a priori to incorporate

the contributions of different unparameterized processes (e.g., the pattern distribution of yellow substances).

BFMCOUPLER solves the sinking processes and is activated for the phytoplankton groups and detritus ($R_{c,n,p}^{(3)}$ variables in Fig. 1):

$$\left.\frac{\partial c}{\partial t}\right|_{bio}^{sink} = w_s \frac{\partial c}{\partial z} \qquad (15)$$

where $w_s$ is the sinking velocity (m/s), which is provided both as a constant value or as a diagnostic result produced by the BFM model based on the nutrient stress conditions of the phytoplankton cells (Lazzari et al., 2012).

A second module of BFMCOUPLER was designed to easily handle the boundary conditions at the surface and bottom. At the surface, air deposition can constitute an important source of nutrients in oligotrophic systems. Furthermore, when the

runoff and nutrient discharge from rivers cannot be incorporated into the MITgcm OBCS package (as in Cossarini et al., 2015a), incorporating these factors as external surface forcings may be necessary (i.e., as localized runoff). Therefore, such contributions are prescribed as additional terms in gchemTendency in the surface layer by reading time-varying 2D maps from external files:

$$\left.\frac{\partial c}{\partial t}\right|_{bio}^{surf} = flux_C|_{surf} \qquad (16)$$



The coupled MITgcm-BFM model includes a simple parameterization of the fluxes at the water-sediment interface, which includes the burial of detritus (e.g., a net export flux from the ecosystem) and an incoming flux of nutrients into the deepest cell of the water column. Burial is parameterized as the first-order kinetics of the carbon (C), nitrogen (N) and phosphorus

(P) contents in the detritus ($R_{c,n,p}^{(3)}$ variables), which sink out of the bottom grid cell:

$$\left.\frac{\partial R_{c,n,p}^{(3)}}{\partial t}\right|_{bio}^{bottom} = -k_{burial} \cdot R_{c,n,p}^{(3)} \tag{16}$$

In the same grid cell, the nutrient (for C equal to $N^{(1)}$, $N^{(2)}$ and $N^{(3)}$ in Fig. 1) bottom fluxes are set either as a constant rate

over the entire domain or as time-varying 2D maps that can be read from an external file or provided by the benthic module of the BFM (which is foreseen in an ongoing development of the model):

$$\left.\frac{\partial C}{\partial t}\right|_{bio}^{bottom} = +flux_C|_{bottom} \tag{17}$$

The BFMCOUPLER involves the use of several external surface forcing fields, such as the surface photosynthetic active radiation, background light extinction factor, sediment fluxes, and partial pressure of atmospheric $CO_2$, which are used by the BFM to calculate the air sea $CO_2$ exchanges. These fields are managed by the BFMCOUPLER package through the BFMCOUPLER_FIELDS_LOAD routine, which is a specifically modified replica of the EXTERNAL_FIELDS_LOAD routine of the MITgcm (Adcroft et al., 2016). This reading of external fields is controlled by two parameters: the period of

forcing (forcingCycle) and the frequency of external forcing (forcingPeriod), which are specified in the BFMCOUPLER namelist (additional details in the Appendix).

### 2.3.3 Compilation and set up

The MITgcm and BFM must be compiled with the same compiler. We tested the code by using both the GNU and Intel

compilers on several HPC platforms. Here, we report the results obtained by running the model (compiled with Intel) on a Linux cluster. The BFM is compiled as an independent library by using the following option of the BFM makefile: mkmf -p $BFM_LIB, and by configuring the config_BFM.sh compiling bash script with the appropriate compilation options (modules, optimization, and compiler), which is also used for the MITgcm compilation. Then, the build_option file for generating the MITgcm makefile must be modified by adding the path to the BFM compiled library and include files.

Additional details are given in the manual of the BFMCOUPLER package (Appendix A).

### 3 Results

We tested the new coupled hydrodynamic-biogeochemical model against two case studies: an idealized experiment (a cyclonic gyre in a mid-latitude closed domain) and a realistic configuration (central Mediterranean Sea). In the first case

study, which was released along with the code and the manual (https://github.com/gcossarini/BFMCOUPLER), we aimed to test the coherence of the model with the expected dynamics based on theoretical considerations and to test the model's performance under different coupling configurations. The second application was not meant to produce a thoroughly validated description of the dynamics of the area but has been designed to show that the coupled model (once run in a





realistic setup) can be used to investigate a wide range of processes from coastal areas to open ocean. A thorough quantitative validation of the Mediterranean model output and an exploration of the results for analyses on the biogeochemical dynamics in the area are beyond the scope of this paper.

### 3.1 Idealized case study

This experiment was based on a simplified case study that consisted of an idealized domain ($2° \times 2° \times 280$-m closed box) that was forced by steady winds and surface heat (downward long-wave and short-wave radiation) and mass (precipitation) fluxes. The horizontal shear in the surface wind field maintained a permanent cyclonic gyre, whereas the surface heat fluxes acted on the thermohaline properties of the water column, inducing a yearly cycle (summer stratification – winter mixing). This simulation was run for several years to reach steady-state conditions (perpetual year simulation).

### 3.1.1 Numerical configuration

This domain was discretized by adopting a uniform grid spacing ($1/32°$) in the horizontal direction, creating 64 grid cells in both directions. All the peripheral grid points of the bathymetry were land points (closed box), whereas the bottom of the domain was a bowl-shaped pit. In the vertical direction, the model was composed of 30 layers with non-uniform thickness (from 1.5 to 21 m). The time step equalled 300 s. External forcing fields were introduced via the MITgcm-native EXF

package. The meteorological forcing consisted of 9 surface fields, namely, the 2-m air temperature (atemp), 2-m specific humidity (aqh), 10-m zonal and meridional wind (uwind, vwind), precipitation (precip), long- and short-wave incident radiation (lwdown, swdown), air pressure (apressure) and surface runoff (runoff). The wind stress and total heat flux were calculated via standard bulk formulae. The experiment was designed with no open boundaries to verify the mass conservation of chemical elements and simulate the effect of free surface dynamics on the distribution of tracers in the

surface layer, which can be important for certain processes, such as the effects of concentration and dilution on the carbonate system variables. We chose the pre-compilation option, which allows for the presence of mass sources/sinks of fluid in the domain (3-D generalization of the oceanic real-fresh water flux option: #define ALLOW_ADDFLUID). With this option enabled, the net contribution of precipitation, evaporation and runoff can be considered in the total mass budget. In particular, we activated the "exact conservation" of fluid in the free-surface formulation (#define EXACT_CONSERV) so

that the temporal evolution of the free surface height exactly equalled the divergence of the volume transport. We allowed the use of the non-linear free-surface option so that the surface level thickness (hFactors) could vary with time (#define NONLIN_FRSURF). The tests were run by adopting the following runtime options (in the namelist "data") for the free surface formulation and the volume conservation constraints:

```
&PARM01
      implicitFreeSurface=.TRUE.,
      exactConserv=.TRUE.,
      useRealFreshwaterFlux=.TRUE.,
      selectAddFluid=1,
linFSConserveTr=.FALSE.,
      nonlinFreeSurf=4,
      &END
```

When configuring the options for the passive tracers package (PTRACERS), we set the concentrations of the tracers in the

surface     mass     fluxes     (evaporation     minus     precipitation     minus     runoff)     to     always     equal     zero



(PTRACERS_EvPrRn(*tracer_number*)=0.0). We used the same advection scheme (3$^{rd}$ order and direct space-time with a Sweby flux limiter [Sweby, 1984]) for active and passive tracers (*tracer*AdvScheme=33). Biogeochemical variables were initialized with suitable vertical profiles for winter condition all over the domain. The BFMCOUPLER package was configured without external forcing both at surface and at the bottom for nutrients, so that a closed system is simulated and

mass conservation is checked. PAR was converted from short wave radiation, and light extinction factor was calculated considering a background values and the shading effect of phytoplankton groups. All details of this experiment along with namelists and input files are given in the Appendix A.

### 3.1.2 Results of the simulation

The model simulated a realistic cyclonic circulation with associated mesoscale variability from vertical thermohaline

stratification and flow instability. Relatively well-mixed thermohaline conditions in the winter induced a more unstable cyclonic gyre with small-scale mesoscale eddies (Fig. 4a), whereas a more stable and energetic cyclonic circulation occurred from stratified thermohaline conditions in the summer (Fig. 4b).

Figure 5 shows the evolution of several physical properties and biological components within the central part of the gyre. The coupled model simulated the evolution of the thermocline and nutricline and the effect of winter vertical mixing on the

temperature and nutrient profiles (Fig. 5a and b). Figure 5 also shows the formation of surface phytoplankton blooms during early winter (Fig. 5c), the formation of the deep chlorophyll maximum (DCM) during summer (as a trade-off between the light penetration and the depth of the nutricline), and the effect of the erosion of the stratification during autumn on the biogeochemical properties of the basin (deepening of mixing layer depth - MLD - Fig. 5a). Net primary production (NPP, contour plot in Fig. 5d) showed the highest values in the proximity of the DCM during spring, although high primary

productivity was also simulated in the upper part of the water column, where the high level of irradiance stimulated carbon fixation, especially for small-sized phytoplankton groups (not shown), even in the presence of low phytoplankton biomass. The region close to the DCM was the most active biological area, i.e., the concentrations of all of the living variables (small and mesozooplankton groups and bacteria; Figs. 5e and f) were the highest and the fluxes fuelled the so-called classic food chain (Legendre and Rassoulzadegan, 1995). Nevertheless, significant bacterial biomass was also simulated in the upper part

of the water column, where bacteria consumed the labile organic matter, which was side-produced by phytoplankton in the well-lighted upper levels. Small zooplankton (sum of micro- and heterotrophic nanoflagellate groups) took advantage of the bacterial biomass, triggering the so-called microbial food web (Legendre and Rassoulzadegan, 1995), which dominated the upper part of the water column during summer. Oxygen (Fig. 5d) was higher in the upper part of the water column during winter because of the high level of NPP and the effect of re-aeration processes with the atmosphere. Bacterial production and

the predominance of respiration over phytoplankton photosynthesis caused the autumn minimum.

### 3.1.3 Application of the longstep option

The computational cost of a 1-year simulation was approximately five hours when adopting an MPI configuration that featured 16 Ivy-Bridge cores. The code profiling (Fig. 6) indicated that most of the CPU time (i.e., up to 85%) was devoted to solving the differential equation for the high number of tracers (51). Solving the transport part (tracer trsp) of tracer

equation (Eq. 10) accounted for 50% of the overall computational cost, whereas solving the biological part (tracer bio) accounted for 35%. The cost of solving tracer transport increased linearly with the number of tracers, whereas the cost of the BFM calculations was primarily dependent on the solution of the carbonate system, although the complexity of the relationships among the biogeochemical variables (results not shown) was also a factor. The use of the MITgcm package LONGSTEP caused an almost exponential reduction in the computational cost for the integration of the tracer equation (Fig.

6). With a Dt$_{trc}$ set to 2400 s (45 min, LSn=8), the runtime was reduced by more than 80% with respect to the reference run.





Assuming this optimization, the fraction that was devoted to the solution of the hydrodynamic and MPI routines accounted for 45%, whereas the remaining part (55%) was devoted to solving the transport-biogeochemical part. Within the tracer equation, 60% of the quota was allotted for transport and 40% of the quota was allotted for the BFM and the other biogeochemical processes.

The use of a coarser time resolution for the solution of the tracer equations implied errors with respect to the reference solution (Fig. 6). The errors were calculated as the root mean square of the difference of the integrated 0- to 200-m chlorophyll between the reference run (LSn=1, i.e., no LONGSTEP) and the run with increased LSn. The magnitude of the mean annual error increased almost linearly with the coarsening of dt and equalled 0.0025 mg/m$^3$ at LSn=8. Within a simulation, the largest differences between the reference run and the coarser time discretization run were registered during

periods with the highest chlorophyll tendency, such as during autumn vertical mixing events along the entire water column and during the deep chlorophyll maximum formation in the spring (not shown). The errors became relevant (>0.01 mg/m$^3$) when larger values for LSn (e.g., LSn>=10) were adopted.

### 3.1.4 Mass budget

The reference run was also used to verify the mass conservation of the coupled hydrodynamic-biogeochemical model by

considering that the model configuration (i.e., non-linear free surface) was set to properly simulate the effects of free surface dynamics on the concentrations of the biogeochemical variables at the surface. Figure 7 shows the time series of the sea level height (eta) averaged over the entire basin. The results indicated the prevalence of rain over evaporation for the first part of the year and vice versa from May to October. For example, the evolution of variable alkalinity, which is a key parameter for resolving carbonate systems in oceans (Follows et al., 2006), within the surface layer was correctly anti-correlated with the

derivative of the eta because the effects of concentration and dilution at the surface are dependent on the water mass balance. This model feature was provided along with the mass conservation capability for tracers (Fig. 7). The errors in mass conservation over time were small ($O(10^{-7})$) and caused by the floating point precision (single) that was used to save the 15-day average model output. The coupled MITgcm-BFM model, which was configured with the non-linear free surface option, allowed us to efficiently simulate the dilution-concentration dynamics while preserving the ability to calculate the budget of

the chemical elements with a high level of accuracy. This feature is indeed important considering the dynamics of variables like alkalinity, whose spatial patterns at the surface were dominated by the regional-spatial-scale distribution of the water mass budget (Cossarini et al., 2015b).

### 3.2 Adriatic-Ionian system case study

The coupled model was also used to simulate a realistic domain: the central Mediterranean Sea. This area, which

encompasses the Adriatic and Ionian Seas (Fig. 8), was chosen because it is characterized by a wide range of interconnected ecosystems that span coastal areas, which are influenced by river discharges, and offshore regions, which are characterized by open-sea dynamics. Indeed, the northern part of the Adriatic is a continental shelf area influenced by terrestrial input (Solidoro et al., 2009, Cossarini et al., 2015a). This area is a site of dense water formation (Gačić et al., 2001, Querin et al., 2013) and represents one of the most productive areas of the Mediterranean Sea (Mangoni et al., 2008). The southern

Adriatic Sea is characterized by an almost permanent geostrophic gyre modulated by deep winter mixing episodes (Gačić et al., 2002, Bensi et al., 2014), and it is connected to the eastern Mediterranean via the Otranto Strait. The Ionian Sea is the deepest sub-basin of the Mediterranean, and it is characterized by basin-scale circulation patterns and smaller mesoscale eddies. This sea is influenced by oligotrophic and salty waters originating from the Levantine basin and by the relatively fresh Atlantic water masses that flow from the west. The hydrodynamics of the area have been simulated by the Adriatic-

Ionian implementation of the MITgcm (ADriatic IOnian System model (ADIOS), Querin et al., 2016), which we used in this





study. The aim of this experiment is to show the ability of the new coupled model to properly simulate the effects of hydrodynamics on biogeochemistry within a wide range of oceanographic and ecological processes that span from a few kilometres to hundreds of kilometres and from oligotrophic to high-level trophic conditions.

### 3.2.1 Domain and model setup

The model domain was delimited by the Sicily channel (Lon 12.2 E) on the western side and by the Cretan Passage (Lon 22.7 E) on the eastern side. The Strait of Messina and the Gulf of Corinth were excluded in this study. The horizontal resolution was 1/32° (approximately 3 km), whereas the vertical grid consisted of 72 z-levels; therefore, the ADIOS model could be easily nested into the 1/16° Copernicus Mediterranean Modelling Forecasting system (CMEMS MED-MFC; Lazzari et al., 2010), which shares the same bathymetry along the open boundary of ADIOS.

The model setup only considered the main rivers that flow into the Adriatic Sea, whereas the minor contributions that flow into the Ionian Sea were neglected. River contributions were introduced as local boundary conditions, imposing observed daily flow rates for the major rivers (e.g., Po) and climatological annual flow rates for the others, with spring and autumn maxima and winter and summer minima (Querin et al., 2013, Janeković et al., 2014). The tracer concentrations at the river mouths were constant in space and time (Table 1), and the mass fluxes were calculated by multiplying the concentrations by

the flow rate of each river.

The boundary conditions along the Sicily Channel and along the Cretan Passage were derived from the CMEMS MED-MFC system (Tonani et al., 2008, Lazzari et al., 2010) for both the hydrodynamic and biogeochemical variables (OBC and $(OBC)_C$ in Fig. 2). The output of the 1999-2012 reanalysis (Salon et al., 2015) was downloaded from the web portal marine.copernicus.eu. The present model configuration adopted a finer horizontal resolution (from 1/16° to 1/32°) with

respect to the CMEMS MED-MFC system, whereas the vertical spacing was the same; hence, interpolating/extrapolating the hydrodynamic and biogeochemical fields in the vertical direction was unnecessary. Furthermore, both the CMEMS MED-MFC system and ADIOS adopted the BFM biogeochemical model; therefore, changes or conversions to the biogeochemical variables were not required. The initial conditions for the hydrodynamic and biogeochemical variables were also derived from the CMEMS MED-MFC system by linearly interpolating the original fields from 1/16° to 1/32°. Additional details on

the ADIOS model setup are provided by Querin et al. (2016).

Surface meteorological forcing was derived from the Regional Climate Model (RegCM) developed at the International Centre for Theoretical Physics (ICTP) in Trieste. We used the 12-km horizontal resolution version with 3-hr output frequency (as in Querin et al., 2016). The heat fluxes ($Q_{sw}$, $Q_b$, $Q_h$, and $Q_e$ in Fig. 2) at the air-sea interface were calculated using standard bulk formulae (via the MITgcm native EXF package); the air temperature, specific humidity, precipitation,

incoming radiation and wind speed values were interpolated from the meteorological model; and the sea surface temperature was provided by the oceanographic model. The 3-hr temporal resolution can highlight the daily variability in the physical and biogeochemical properties of the uppermost layers of the water column (daily cycling of the PAR, temporal variability in the temperature and wind).

The specific settings for the BFMCOUPLER package were specified as follows. The background water light extinction

factor was set to 0.035 $m^{-1}$ and the coefficient for the self-shading effect by phytoplankton was $10^{-3}$ $m^2$/mg CHL for diatoms and $8 \times 10^{-3}$ $m^2$/mg CHL for the other three phytoplankton groups. The nutrient surface forcing (air deposition) was set to 0.00096 and 0.057 mmol/$m^2$/d for phosphorus and nitrate, respectively (Lazzari et al., 2012 and reference therein), whereas we assumed that the atmospheric $pCO_2$ linearly increased from 380 to 395 in the period 2006-2012 according to the trend that was reported in Artuso et al. (2009). No bottom forcing was prescribed for the biogeochemistry.





### 3.2.2 Results of the simulation

The simulation covered the period from January 2006 to December 2012 at a time step of 200 s. In the following analysis, we disregarded the first 2 years of the simulation, which we considered a spin-up period for the biogeochemical variables from the CMEMS's coarser resolution fields. The MPI domain decomposition consisted of $24 \times 18$ subdomains run on 224 Intel Xeon Ivy Bridge cores of a Linux cluster, and the computational cost of the simulation was 65.8 hr per year. The runtime was significantly reduced by adopting the LONGSTEP option (Table 2). The wall clock time progressively decreased by increasing the longstep factor (LSn) from 1 to 9. Then, time steps that were higher than 30 min substantially decreased the accuracy without further reducing the computational cost because large oscillations were generated in the solution.

We present the results for the ADIOS case study to demonstrate the ability of the new MITgcm-BFM coupled model to investigate closely interconnected hydrodynamic and biogeochemical processes for both coastal and open sea ecosystems.

On the western coastal areas of the Adriatic Sea, the maps in Figure 9 correctly display the patterns of low salinity, southward currents, high nitrate and chlorophyll concentrations, and strong primary production, which are all typical fingerprints of the Western Adriatic Current (WAC) system in the Adriatic Sea. The effect of the input from the northern rivers and the basin-scale cyclonic circulation generates a frontal system along the Italian coast. As is commonly observed in satellite chlorophyll maps (Barale et al., 2005), the width of the WAC frontal system decreases southwards, whereas weaker recirculation patterns are also visible in the central Adriatic Sea (Fig. 9). Other river-influenced coastal areas are simulated along the south-eastern areas of the Adriatic Sea, where the input from the Neretva and other south-eastern rivers triggers small-scale chlorophyll-a signals along those areas, as reported by Marini et al. (2010). The northward flow of salty and oligotrophic water, which enters through the Otranto Strait, confines the river's fertilization to a narrow coastal strip.

The coastal to open-sea gradients of nutrients were accurately simulated by the coupled model. As an example, Figure 9 shows that the nitrate patterns display a longitudinal gradient along the Adriatic and northern Ionian seas, and these results are consistent with the current climatologies (Cossarini et al., 2012, Solidoro et al., 2009, Zavatarelli et al., 1998). In the open-sea area of the Ionian Sea, the surface circulation is dominated by large mesoscale structures and a basin-scale anticyclone in the middle, and the downwelling area is characterized by minimal nitrate and chlorophyll concentrations (Fig. 9). This pattern is consistent with the climatology of Manca et al. (2004), even if the nitrate concentrations are slightly higher in the eastern Ionian Sea, which is related to overestimated eastern boundary values.

If we focus on the open-sea sub-surface dynamics, we can analyse how vertical processes affect the biogeochemistry. The vertical profiles of chlorophyll and phosphate for the two sites in Fig. 8 are depicted in Fig. 10. One site is located in the centre of the southern Adriatic gyre, which is characterized by strong winter vertical mixing, whereas the second is located in the centre of the large anticyclonic gyre in the Ionian Sea. A comparison between the two sites shows the ability of the coupled model to simulate the different regimes in the two areas. The southern Adriatic Sea presents a much deeper mixed layer depth, a shallower nutricline than the Ionian Sea, more intense inter-annual variability in the cyclic alternation of winter vertical mixing phases, and the onset of summer stratification.

The intense vertical mixing in the southern Adriatic area during winter drives the upwelling of nutrient-rich water, which contributes to a shallow nutricline (up to the depth of the DCM) during summer. However, winter ventilation in the Ionian Sea's open areas rarely reaches a depth of 250 m; consequently, nutrient-rich water remains confined to the deepest layers (below 200 m). The two areas are characterized by different biological regimes because of the different depths of the nutricline and the superimposed longitudinal gradient of the background light extinction factor (according to Lazzari et al., 2012).

Another interesting coupled hydrodynamic-biogeochemical feature is displayed along the southern coast of Sicily, where the entrance of modified Atlantic water (MAW, low-saline water mass in Fig. 9a) and the simulated coastal upwelling from





westerly winds, induce vertical transport of nutrients, consistently with the findings of Patti et al. (2010) and Rinaldi et al. (2014). Intense vertical dynamics trigger the high concentrations of nutrients and chlorophyll and the strong primary production simulated in the upper layer of the northern Sicily channel (Fig. 9b), and these results are consistent with the typical patterns observed in satellite chlorophyll maps (Volpe et al., 2012).

The computation and diagnostics of the transport components for the tracers (e.g., zonal and meridional advection and diffusion, vertical advection and implicit and explicit diffusion) are already implemented in the native PTRACER and DIAGNOSTIC packages of the MITgcm. This feature, which is complemented by the ability to calculate the surface and lateral fluxes at the boundaries through the BFMCOUPLER package, allows us to calculate the budget of the simulated chemical elements in marine ecosystems. As an example, we evaluated the meridional transport across the Otranto Strait for

the carbon components along with other fluxes at the domain interfaces (i.e., the $CO_2$ flux at the air-sea interface and the river input) to calculate the carbon budget in the Adriatic Sea. The results show that the Adriatic Sea acts as a downwelling pump of carbon for the Mediterranean Sea. In particular, the Adriatic Sea imports carbon from rivers (3.17 $10^{12}$ g C/y) and from the atmosphere (1.65 $10^{12}$ g C/y). At the Otranto Strait, the Adriatic Sea imports carbon through the surface layer (0-200 m): 192.7 $10^{12}$ g C/y in terms of DIC and 0.2 $10^{12}$ g C/y in terms of organic carbon. Conversely, this sea exports carbon

through the bottom layer (200-1000 m): 197.7 $10^{15}$ g C/y and 0.03 $10^{15}$ g C/y in term of DIC and organic carbon, respectively. Finally, the Adriatic Sea is a net sink (approximately 4.7 $10^{12}$ g C/y) of carbon into the interior of the Mediterranean Sea. In terms of the transport across the Otranto Strait, Figure 11 shows the complex structure of the northward (red) and southward (blue) fluxes simulated by the coupled model. In particular, organic carbon (sum of all the living components: $P_c^{(1,2,3\ and\ 4)}$, $Z_c^{(1,2,3\ and\ 4)}$ and $B_c^{(1)}$; detritus, $R_c^{(3)}$; and dissolved organic carbon, $R_c^{(1,2\ and\ 4)}$) is mainly

confined to the surface layer for both the inflow and outflow. A barely visible flux of organic carbon toward the Ionian Sea is depicted along the western slope below a depth of 200 m (mainly because of the sinking of detritus). The northward and southward fluxes of DIC along the surface (Fig. 11) are characterized by the same organic carbon pattern and nearly balanced. Additionally, an outflow (blue) area at a depth of 300-900 m along the left flank of the strait indicates DIC transport associated with the Adriatic Dense Water Outflow Current (DWOC, Gačić et al., 2001). This carbon flux

represents the export term that closes the budget of the Adriatic Sea and replenishes the layer of the Ionian Sea below the depth of the Levantine Intermediate water, which suggests a possible mechanism for the long-term carbon sequestration in the Mediterranean Sea.

**4 Discussion and conclusion**

In this paper, we presented a coupling framework between two widely used models, the MITgcm and BFM, and we showed

the potential of the new coupled model. These two models were developed by two different scientific communities that are actively and constantly involved in improving the codes. When one model is directly embedded into another, code developments might represent an issue because of the constant and tedious work of keeping one code updated with respect to the other. Therefore, the coupling framework in this paper was designed to preserve the independence of the two models as much as possible. The number of modifications that were required for the two original codes was limited, and changes could

be easily managed should each single model be upgraded. In our solution, the MITgcm remained the host code, the BFM was compiled and linked as an independent library, and the new BFMCOUPLER package handled all the coupling procedures and concentrated all the coding effort. The upgrades to the MITgcm enumerated less than 10 new code lines in a few routines (in the GCHEM and LONGSTEP packages) and the list of available diagnostics (in the DIAGNOSTIC package). On the BFM side, several "include" files contained a list of newly added variables. The order of the variables in

the BFM's include files and in the MITgcm's file data.ptracer must be consistent (see Appendix A). This feature is important



because the BFM's codeV5 (Vichi et al., 2015) can be customized in terms of both the number of state variables and processes, thus increasing the flexibility of the new coupled model for a wider range of applications.

Several coupling environments are already available (e.g., FABM, Bruggeman and Bolding, 2014; and MESSy, Jockel et al., 2008). However, our coupling scheme fully exploits the parallelization efficiency of the MITgcm through a domain
decomposition algorithm (Marshall et al., 1997). The programming effort required for this feature, which may be crucial for resolving large domains, is not much greater than what is already required to run the MITgcm model alone.

Other biogeochemical models of various complexity have already been embedded into the MITgcm (Dutkiewicz et al., 2009; Hauck et al., 2013; Cossarini et al., 2015a). Nevertheless, the BFM in this new coupled model has a biological complexity and a number of features (Lazzari et al., 2016) that increase the attractiveness of the model for many marine applications.
The MITgcm-BFM coupling scheme was primarily designed by considering the source splitting approach because this framework has the highest level of numerical accuracy (Butenschön et al., 2012). The use of the longstep option reformulated the coupling as a processor-splitting algorithm that allows for different time steps for hydrodynamics and coupled transport-biogeochemistry at the cost of accuracy. When using the longstep option, the results (Fig. 6 and Table 2) show that the loss of accuracy remained negligible only for a limited increase in the tracer time step. Furthermore, the
coupling framework could handle a separate solution of hydrodynamics and transport processes from the biogeochemical processes through the use of the gchem_separate_forcing option (Fig. 3). However, this approach would require a wider modification of the gchem package to introduce independent integration steps for the transport and biogeochemical parts of the tracers. Then, a more detailed analysis of the sensitivity (e.g., similar to what was proposed in Butenschön et al., 2012) of the biogeochemical model's results to the different coupling schemes and time steps should be performed for each specific
application.

A source-splitting scheme might be more appropriate for investigating the feedback of the biogeochemistry on the hydrodynamics of the system. An example is the calculation for the sinking of certain phytoplankton groups, which is a physical 1D process solved within BFMCOUPLER and related to the sinking velocity calculated by the BFM. Furthermore, the shading effect on light penetration caused by phytoplankton and other suspended matter currently only affects the PAR
vertical profile (Eq. 14). However, this factor could be introduced as an extra term in the routine that calculates seawater thermodynamics (in the routines SWFRAC and EXTERNAL_FORCING). A new parameterization of the penetration of solar radiation could be used to estimate the biological effects on the seawater temperature, which might be an interesting issue in highly productive areas, such as the northern Adriatic Sea and the coastal strip along the Italian coast reached by the Western Adriatic Current (WAC). A realistic simulation of light absorption with depth could reduce the model errors when
estimating temperature, which is affected by many other sources of uncertainty from the surface forcing data, the heat flux bulk formulation, the vertical resolution and the parameterization of vertical turbulent processes. The design of our coupler, which is characterized by the sharing of biogeochemical variables and their tendencies in the host model's memory structure, allows for the future implementation of the feedback effects of biology on hydrodynamics.

Furthermore, the new coupling scheme was designed to foster development towards a full Earth system modelling approach,
in which a wide range of processes among the Earth's spheres can be simulated online and the interactions and feedback effects can be directly considered. For example, the BFM has already been coupled with other ecosystem components (e.g., online coupling with the high-trophic-level model Ecopath with Ecosym, Akoglu et al., 2015). Moreover, the parameterization of Eq. (16) and (17) can be easily substituted by a call to a benthic model function, which solves the processes that occur in a single-layer sediment model and calculates the exchanges between the pelagic environment and the
sediment.

Similarly, the MITgcm has already been coupled with atmospheric models. For example, the MITgcm has been coupled online with the atmospheric model RegCM in the Mediterranean Sea region (Giorgi et al., 2006) using the coupling



framework OASIS (Artale et al., 2010). Therefore, our coupling scheme can act as a link between atmosphere-hydrosphere models and biosphere models. This coupler could be successfully used to study ocean-atmosphere interactions, such as the effects of climate scenarios on high-trophic-level ecosystem components or the feedback of ocean carbon pumps on the climate.

Finally, the results of the two test cases show that the new coupled model provides a realistic representation of a wide range of marine processes from costal to open-sea ecosystems, where the interplay of hydrodynamics and biogeochemistry is crucial. The effects of river plumes, coastal upwelling, and different vertical mixing regimes on phytoplankton dynamics were reasonably reproduced by the model and found to be consistent with both theoretical knowledge (Mann and Lazier, 2006) and published experimental findings for the Mediterranean Sea.

**Acknowledgements**

This study was supported by the Italian flagship project RITMARE.

**Appendix A: Manual for the implementation and use of the BFMCOUPLER package**

**A.1 Introduction**

This package was developed as a specific interface among the MITgcm, the GCHEM package and the Biogeochemical Flux

Model (BFM). The BFM (bfm-community.eu) is a complex and modular biogeochemical model that was designed to simulate multiple plankton functional types and the cycling of several chemicals (i.e., carbon, nitrogen, phosphorus, silica, and iron) within the marine pelagic ecosystem. BFMCOUPLER was designed to handle the input/output directives between the MITgcm and BFM and to reproduce several processes (light extinction, sinking, and biogeochemical chemical fluxes at the air-sea and sea-bottom interfaces) that are not considered in both models. For more details regarding the equations, see

section 2 of this paper.

**A.1.1 General architecture of the coupled model**

Several hydrodynamic-biogeochemical coupling options were implemented according to a previously implemented option in the GCHEM package. The gchem_separate_forcing option controls how and when the tracer tendencies are calculated and applied. The use of the LONGSTEP package is another coupling option available with BFMCOUPLER.

**A.2 Key subroutines and parameters**

**A.2.1 Initialization**

BFMCOUPLER_VARS.h contains the common blocks for the list of the BFM's state variables and diagnostic variables (BFM_var_list.h) and for the parameters and fields that are required to calculate the carbonate system solution, $CO_2$ air-sea exchange, PAR, light extinction, sinking and nutrient air deposition and bottom fluxes. Forcing fields can be initialized

either with a background value by BFMCOUPLER_INI_FORCING.F or read from external fields. BFMCOUPLER_READPARAMS.F reads the namelist data.bfmcoupler, which contains the names of the files for the above fields. The parameters that manage the time intervals for reading, interpolating and applying the external forcings are read from the above namelist. The input namelist also contains specific parameters for the processes solved by BFMCOUPLER: sinking speed for detritus, self-shading coefficients for different phytoplankton groups, and background values of the

seawater light extinction factor. The allocation of memory used by the BFM is set here by the BFM routine BFM_initialize.




### A.2.2 Forcings

The advection-diffusion tendencies of tracers are calculated in ptracers_integrate.F, whereas the biogeochemical process tendencies are handled by the routine BFMCOUPLER_CALC_TENDENCY.F, which is called from the opportunely modified GCHEM_CALC_TENDENCY.F, and controls the following:

- input directives to the BFM routine BFM_input_ecology for the tracer values and all the necessary information used by the BFM itself (coordinates of the cells within the water column, temperature, salinity, PAR, atmospheric $CO_2$ pressure, and wind speed in the corresponding surface grid point);

- call to the BFM model (BFM0D_ecology_dynamics);

- calculation of the PAR, the sinking of phytoplankton and detritus, and the atmospheric deposition of nutrients and bottom

fluxes;

- output directives from the BFM routine BFM_output_ecology for transferring and applying biogeochemical tendencies and diagnostics.

### A.2.3 Loading fields

The external forcing fields used by the BFMCOUPLER (e.g., $CO_2$ air concentration, PAR, light extinction factor, nutrient air

deposition, and bottom fluxes) are read by the routine BFMCOUPLER_FIELDS_LOAD.F, which is called from the opportunely modified GCHEM_FIELDS_LOAD.F.

### A.2.4 Diagnostics

The BFMCOUPLER package uses the MITgcm's DIAGNOSTICS package. The definition of new specific diagnostics from the BFM's fluxes and variables is managed in BFMCOUPLER_DIAGNOSTICS_INIT.F, which is called from

BFMCOUPLER_INIT_FIXED.F. The new diagnostics quantities are calculated in BFMCOUPLER_CALC_TENDENCY.F through a list of files (BFMcoupler_VARDIAGlocal.h, BFMcoupler_VARDIAGcopy_fromD.h and BFMcoupler_VARDIAG_fill_diags.h) that use the variables from the BFM routine BFM0D_output_ecology and specific instructions from the diagnostics package (DIAGNOSTICS_FILL.F routine).

New diagnostic quantities are listed in the namelist in the parameter file data.diagnostics, which specifies the frequency and

type of output, the number of levels, and the names of all the separate output files.

The coupled MITgcm-BFM model can use a large number of tracers; therefore, increasing the ndiagMax parameter in diagnostics_size.h may be necessary.

### A.2.5 LongStep

The MITgcm package LONGSTEP allows the tracer time step to be longer than the time step used by the hydrodynamic

model. When this package is activated along with the BFMCOUPLER package, a new specifically developed version of the routine LONGSTEP_THERMODYNAMICS.F has to be used. The new version of this routine includes a call to BFMCOUPLER_CALC_TENDENCY. The BFMCOUPLER routines use the hydrodynamic variables stored in the LONGSTEP variables, which are either the averages or temporal sub-samplings of the variables of the master hydrodynamic model depending on the when_to_sample parameter set in the data.longstep namelist file.

### A.2.6 Compilation and compile time flags

The BFM is a Fortran95 code and must be compiled separately as an external library in advance ($BFM_LIB/lib/libbfm.a). According to the BFM's manual, a compiled library version is obtained by customizing the BFM makefile (mkmf -p




$BFM_LIB). The config_BFM.sh compiling bash script must contain build options (modules, optimization options, and compiler) that are consistent with those of the MITgcm compilation.

When the MITgcm is compiled, the build_option file must be modified and the following lines must be added:

BFM_LIB=$BFM_PATH/lib

BFM_INC=$BFM_PATH /include

export LIBS="$LIBS" –L $BFM_PATH/lib –lbfm

export INCLUDES="$INCLUDES -I$BFM_PATH /include"

The subroutines of the new package BFMCOUPLER must be included in the folder

/MITgcm/pkg/BFMCOUPLER,

which can be added to the original source tree of the code. BFMCOUPLER must be specified in the compile configuration file packages_conf.

Several specific compile time flags are set in BFMcoupler_OPTIONS.h:

USE_QSW: use Qsw from the MITgcm to calculate the photosynthetic active radiation (PAR)

READ_PAR: read the PAR from a file set in data.bfmcoupler.

USE_SHADE: include the role of phytoplankton and detritus in the calculation of the vertical profile of the PAR.

READ_xESP: read the background light extinction factor from a file set in data.bfmcoupler.

USE_SINK: use the calculation for the sinking of phytoplankton and detritus.

USE_BURIAL: calculate the contribution of burial for detritus tendency at the bottom.

USE_BOT_FLUX: use input sediment fluxes for nutrients at the bottom.

BFMCOUPLER_DEBUG: activate a control on the tendencies calculated by the BFM.

**A.3 Do's and Don'ts**

This package must be run with both PTRACERS and GCHEM enabled. This package is configured for a number of biogeochemical variables specified by the BFM model. Therefore, data.ptracers must be configured accordingly (order of tracers equals what is specified in the ModuleMem.F90 file from the BFM code). This package must also be run with
diagnostics enabled.

**A.4 Code availability and the experiment that uses BFMCOUPLER**

The code can be downloaded from the https://github.com/gcossarini/BFMCOUPLER link.

The numerical experiment described in this paper (sec. 3.1) consists of an idealized domain forced by steady wind and surface heat and mass fluxes. This case study simulates a permanent cyclonic gyre with a yearly cycle of thermohaline and
biogeochemical properties. The input files along with the MITgcm and BFM namelists of the experiment are available at the https://github.com/gcossarini/BFMCOUPLER/input/ link.

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



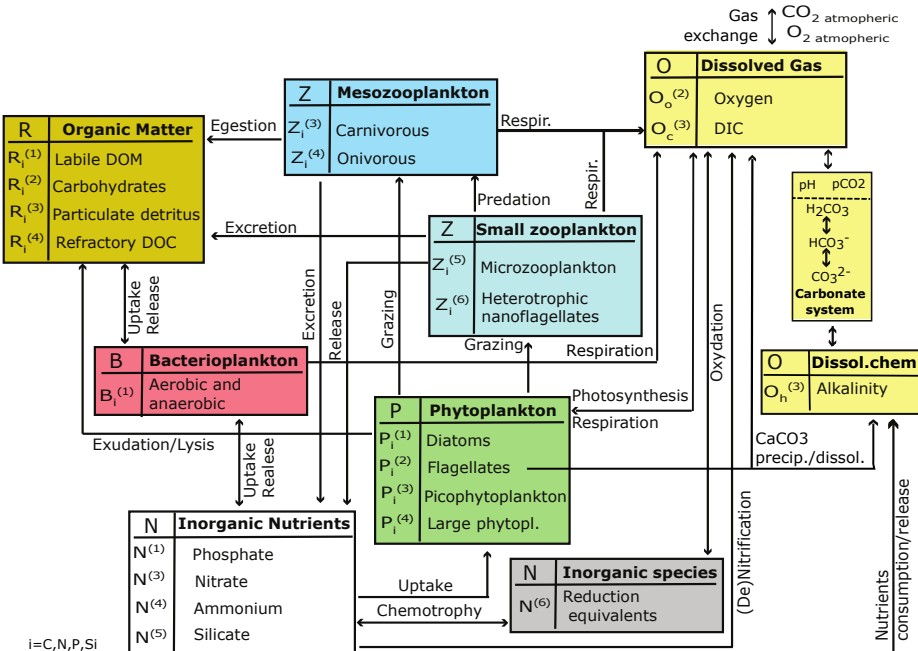

**Figure 1: BFM model: scheme of the functional interactions among the variables in the version that was implemented in Lazzari et al. (2012), Melaku Canu et al. (2015), and Cossarini et al. (2015b).**

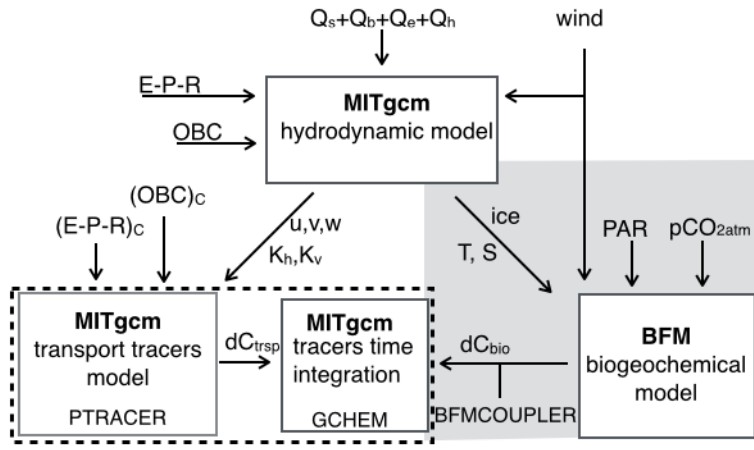

**Figure 2: Description of the MITgcm-BFM coupling and interfaces among the different components of the coupled model. $Q_{sw}+Q_b+Q_e+Q_h$ (sum of shortwave radiation, backward radiation, latent and sensible heat fluxes) represents the total heat flux ($Q_\theta$ in Eq. (5)).**



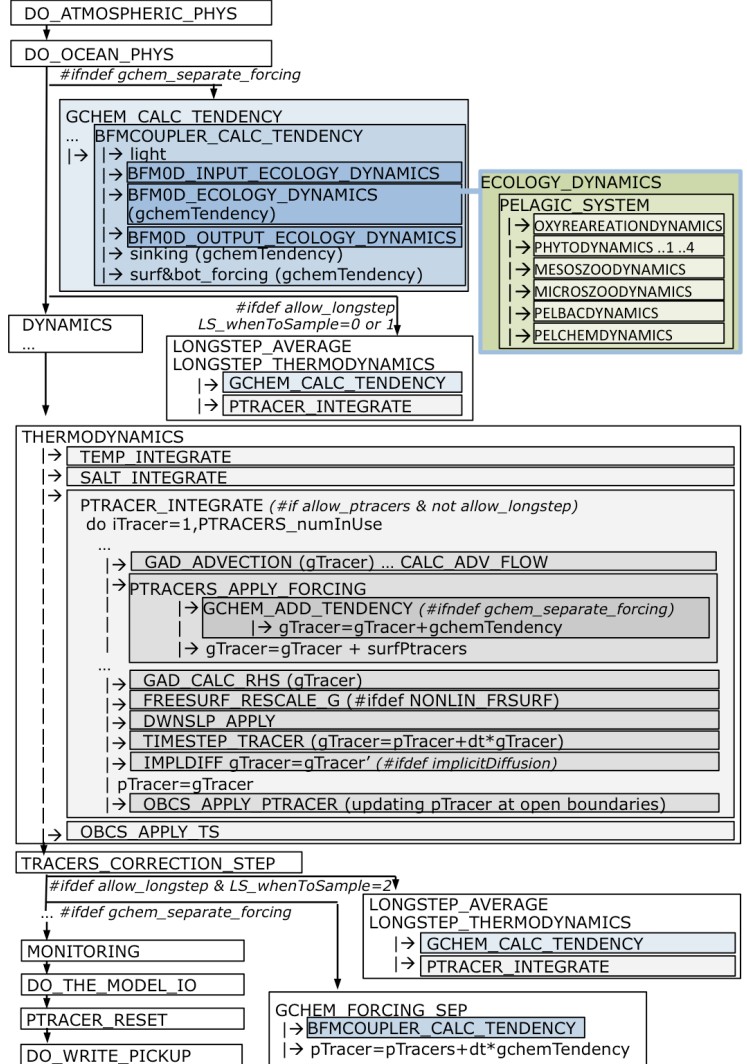

**Figure 3: Workflow of the MITgcm routine FORWARD_STEP. The boxes indicate the routines and their dependencies. The matrix of the tracers' state variables (pTracer), the overall tendency of the tracer (gtracer), and the tendency for the biogeochemistry only (gchemTendency) are also specified. The blue boxes indicate modifications to either the MITgcm code or the BFMCOUPLER routines, whereas the green boxes indicate BFM routines. The pre-compilation options (#) and omitted parts (…) are also shown.**



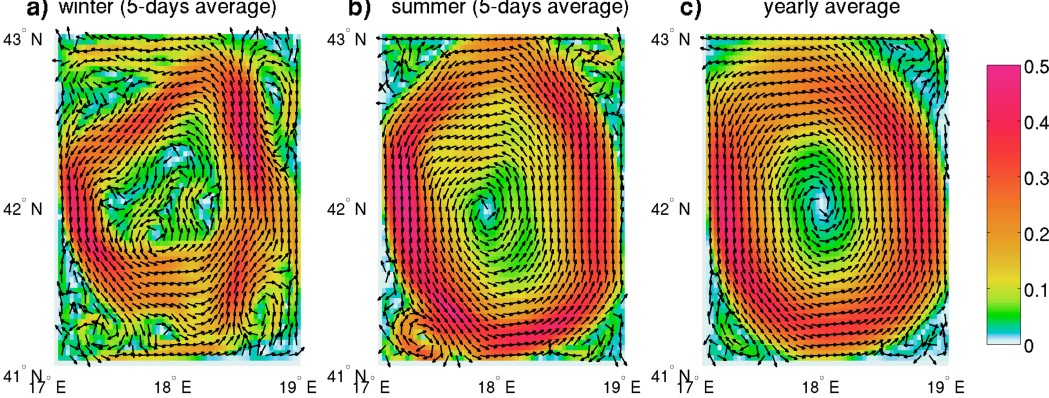

**Figure 4: Current speed (colour, m/s) and direction (vectors) at 12-m depth: 5-days average in winter (a) and summer (b), and yearly average (c).**

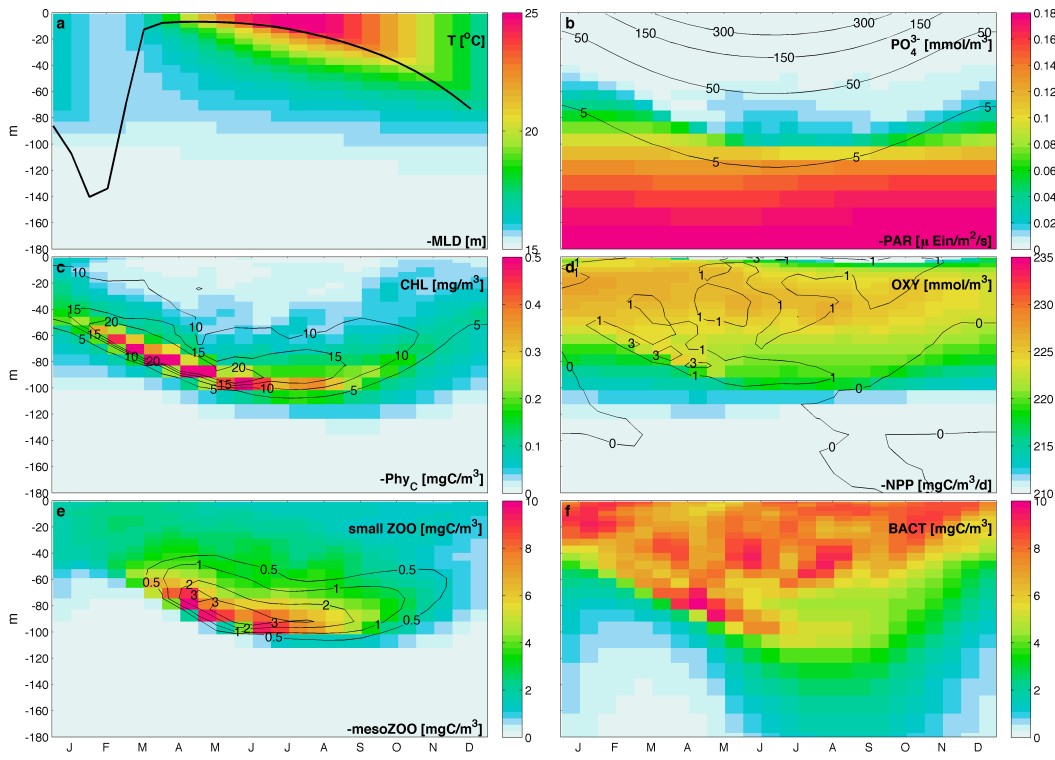

**Figure 5: Howmöller diagrams of the (a) temperature and evolution of the mixed layer depth (contour), (b) phosphate and PAR (contour), (c) total chlorophyll (sum of the chlorophyll content in the four phytoplankton functional groups) and total**
10 **phytoplankton biomass (Phy_C - contour), (d) oxygen and NPP (contour), (e) small zooplankton and mesozooplankton (contour), and (f) bacteria.**

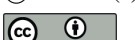



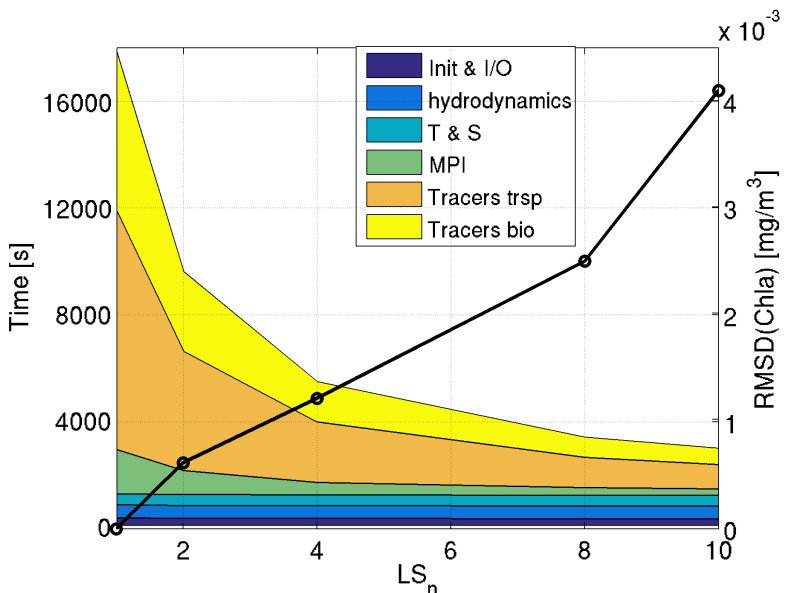

**Figure 6: Wall clock time of the simulations as a function of LSn for selected parts of the model (left axes): Init & I/O (MODEL_I/O, DO_STATEVARS_DIAGS, LOAD_FIELDS_DRIVER, MONITOR, DO_THE_MODEL_IO and DO_WRITE_PICKUP), hydrodynamics (DYNAMICS, SOLVE_FOR_PRESSURE, INTEGR_CONTINUITY and other routines); T & S (TEMP_INTEGRATE and SALT_INTEGRATE); MPI (BLOCKING_EXCHANGES); Tracers bio (GCHEM_CALC_TENDENCY) and trsp (PTRACER_INTEGRATE). The root mean square error of the integrated chlorophyll (right axis) is shown as a function of LSn.**

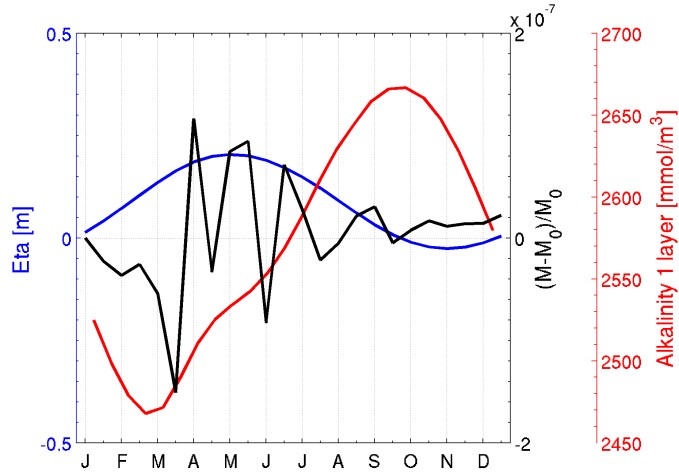

**Figure 7: Evolution of Eta (blue line) and alkalinity concentration (red line) at the surface layer together with the relative mass variation with respect to the initial conditions (black line). The total alkalinity mass was obtained by multiplying the 15-day average model output (saved with single floating point precision) by the domain volume, which included the time-varying eta at the surface layer.**




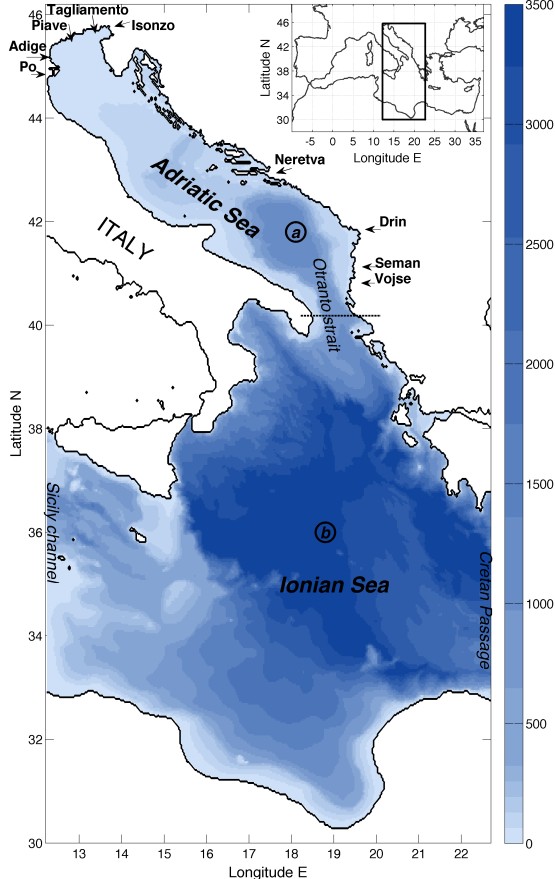

**Figure 8:** Bathymetry (depth in meters) of the Adriatic-Ionian model. The plot also indicates the location of the major rivers (arrows), the Otranto Strait and the position of the 2 sites (circles) that were selected to display the Howmöller diagrams in Fig. 10.



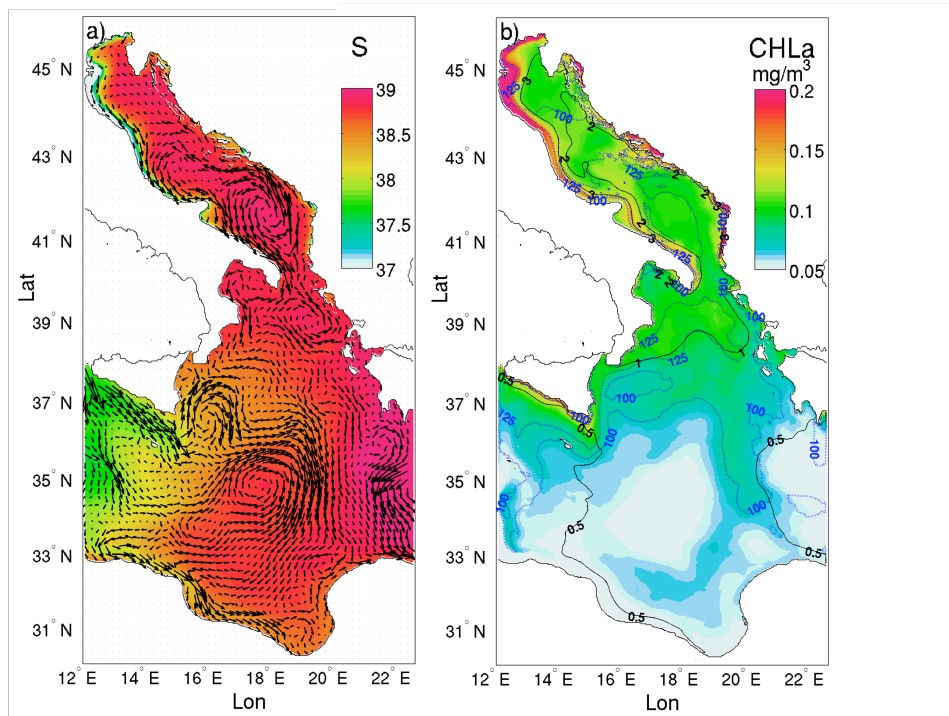

**Figure 9: Map of the surface currents (arrows) and salinity (a). Map of the surface chlorophyll (b) and contours (solid black lines) of nitrate concentration [mmol/m3] in the upper layer (0-20 m), and contours (dotted blue lines) of the annually averaged and vertically integrated (0-200 m) net primary production [g C/m2/y].**

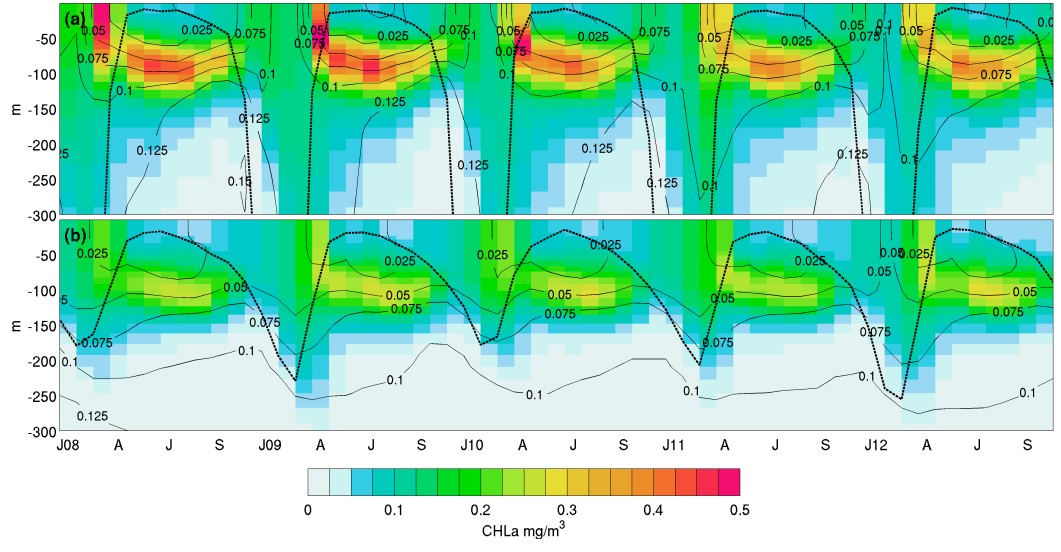

**Figure 10: Howmöller diagrams of chlorophyll and phosphate and plots of the mixed layer depth (dashed lines) for the southern Adriatic Pit (a) and the Ionian offshore area (b).**



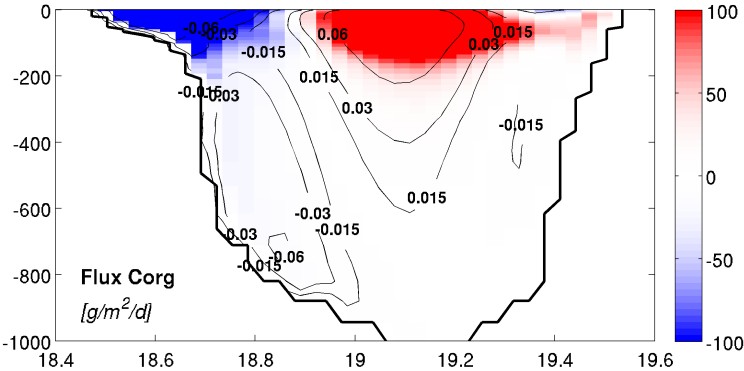

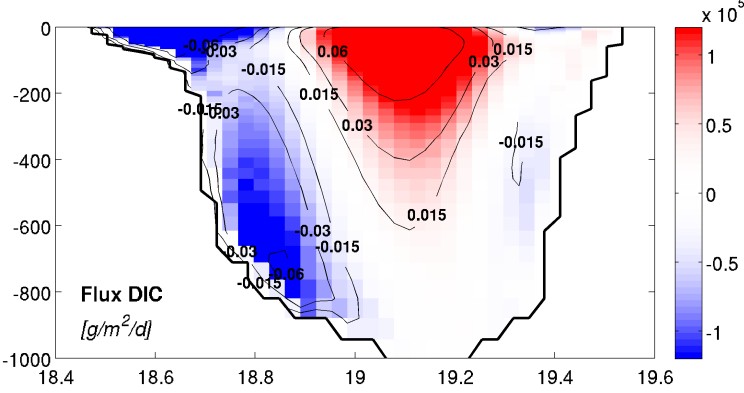

**Figure 11: Fluxes of Corg (upper panel) and DIC (lower panel) across the Otranto Strait (dashed line in Fig. 8). The solid contours specify meridional velocities (with positive velocities oriented northwards).**



| BFM name (Fig. 1) | Variable name | Unit | value | reference |
|---|---|---|---|---|
| O[(2)] | $O_2$ | $mmolO_2/m^3$ | 250 | Saturation level in fresh water |
| N[(1)] | $PO_4^{3-}$ | $mmolP/m^3$ | 2.6 | Cossarini et al., 2015a, adapted from Ludwig et al., 2009 |
| N[(3)] | $NO_3^-$ | $mmolN/m^3$ | 150 | Cossarini et al., 2015a, adapted from Ludwig et al., 2009 |
| N[(4)] | $NH_4^+$ | $mmolN/m^3$ | 34.1 | 1/5 of total N ($NO_3^-$ + $NH_4^+$) |
| N[(5)] | $SiO_2$ | $mmolSi/m^3$ | 150 | Set equal to N3n value$^-$ |
| $O_c$[(3)] | Dissolved Inorganic Carbon (DIC) | $mgC/m^3$ | 33225 | Cossarini et al., 2015b |
| $O_h$[(3)] | Alkalinity | $mmol/m^3$ | 2800 | Cossarini et al., 2015b |

5   **Table 1. Tracer concentrations in the rivers.**

| LSn | 1 (Ref) | 3 | 6 | 9 | 12 |
|---|---|---|---|---|---|
| **$Dt_{trc}$ (s)** | 200 | 600 | 1200 | 1800 | 2400 |
| **Wallclock time (h) per 1-year simulation** | 65.8 | 29.5 | 17.3 | 14.5 | 15.1 |
| **Error of integrated 0-200m chlorophyll** | 0 | 0.01% | 0.05% | 0.1% | >10% |

**Table 2. Computational cost as a function of the factor LSn and the mean error of the integrated chlorophyll. The error was the annual average of the RMS error of the differences between the longstep and the reference (Ref) simulations. The error was**
15   **normalized using the reference simulation.**