# Peer review of "Development of BFMCOUPLER (v1.0), the coupling scheme that links the MITgcm and BFM models for ocean biogeochemistry simulations"

_Geoscientific Model Development, 2016_

## Referee Comment (RC1) · Anonymous Referee #1 · 23 Dec 2016

**1 Referee report "Development of BFMCOUPLER (v1.0), the coupling scheme that links the MITgcm and BFM models for ocean biogeochemistry simulations" by Cossarini et al. Geosci. Model Dev. Discuss., 2016**

**1.1 General Remarks**

This manuscript presents a software interface that links two well established models within the marine science community for ocean circulation and marine biogeochem-

istry, providing a new valuable tool for a vast range of applications indicated in the work. I'd like to compliment the authors for their effort in covering the wide range arguments involved in describing a coupled hydrodynamic biogeochemical system comprehensivley in a reasonable amount of space with an adequate level of detail without any major omission. The work covers all relevant scientific and technical aspects and is therefore certainly eligible for publication in Geoscientific Model Development after that some minor issues have been addressed which concern some missing technical information and inprecsions in the scientific description, that are given below.

**1.2 Technical Omissions**

In order to benefit users of this software interface, I believe the addition of the following informations would be crucial:

- The version of the MITgcm used in this work has been specified, but an indication of where to retrieve the model code from would be helpful. Of course, not being the developers themselves, the authors will have no control of the future accessibility to the point of retrieval, but at least an indication of where or how to obtain the code at present is required.

- What is the precise version of the BFM code, that was used? I gather the model is distributed via git, so a commit hash and the link to the repository would suffice.

- Programming languages and version of all parts should be specified. To my understanding all three are coded in FORTRAN, but only for the BFM this is clearly stated including the FORTRAN version.

- As I assume develpoment of the couple will continue, also a hash for the coupler library is needed.

- Specifications of input/output formats are missing.

**1.3 Detailed Comments**

**page 2, line 12:** Check author names and publication year.

**page 2, line 15:** I don't think it's a matter of complexity, but a structural argument that dictates the model hierarchy between physical driver and biogeochemical model: in all cited cases and all other couplings I am aware of, the physcial model provides the overarching geometrical structure of the model (and can run on its own), while the biogeochemical model typically just computes biogeochemical rates on a per pixel base and at the bare minimum requires a 0D driver to run any simulation. In fact, I don't think that the statement that the physical models are far more complex holds generally.

**page 2, line 18:** Instead of "input/output directives" I'd prefer interface specifications or if you want more use the technical term application programming interfaces (APIs) in order to not confuse with actual model in- and output data.

**page 2, line 32-34:** The choice of the authors in itself doesn't require justification, but is there any evidence that performance would suffer from using these tools? I would have thought that depends (to some degree) on how well they are written?

**page 2, lines 34-38:** Worth mentioning here if it deals with online, offline coupling or both.

**page 4, line 36:** Any explicit time integration method is forward-in-time, that includes Adams-Bashforth schemes. Which one is used here, Euler forward?

**equation 10:** The meaning of the hereto unused variables needs to be specified. Specifically the meaning of the two diffusion terms should be made clear. Also, as these equations are still generic at this point, it may be worth using a different sub-script for the biogeochemical sources and sinks.

**page 4, lines 9-15:** Might be worth also to refer already to the possibility of surface/bottom boundary conditions for the biogeochemistry introduced in the BFM-COUPLER section later.

**page 6, lines 16-17:** The information flow between BFM, MITgcm and the coupler in between is not one-way as clearly indicated by figure 2 and the following text. It is true that the ocean physics in the described setting remain unaffected by the biogeochemistry, but that doesn't mean that the information flow is one-way, as the transport model for passive tracers sits in the MITgcm code and requires the biogeochemical sources and sinks for integration (and biogeochemical vertical movement).

**page 6, line 33:** "a source splitting scheme is adopted": the insertion of the biogeochemical rates into the transport solver constitutes a source splitting only of the integration scheme apllied effectively omits information or intermediate steps that are required by the coupler. In particular, if the integration scheme is a simple Euler forward scheme, the insertion of the biogeochemical rates does not involve any form of splitting, but constitutes a direct integration of the terms in equation 10.

**page 7, lines 23-24:** The option of applying different time steps to the two modes is not unique to the process splitting, but can be easily adopted in the source spliting method by updating the rates of the slower process only on intermediate time steps. (E.g. Blom, J.G., Verwer, J.G., 2000. A comparison of integration methods for atmospheric transport-chemistry problems. Journal of Computational and Applied Mathematics 126, 381–396. doi:10.1016/S0377-0427(99)00366-0)

**page 7, line 26-27:** "to increase the computational performance of the entire code" In which way?

**page 8, line 15 ff, eq 14:** What would actually be required is the average light in each

cell, which given the exponential distribution is not the light in the cell centre. Using the same formula, it is straight forward to compute the integral of light between the upper and lower cell face and divide by the cell thickness in order to arrive at the correct number.

**page 8, eq. 15:** How is this pde solved numerically? Provide the scheme or a reference to a full descrition.

**page 9, line 5:** What happens to sinking phytoplankton that hits the sea floor?

**page 9, line 28:** ". . . which are . . . "

**page 10, line 6:** Further down it appears that at least the surface heat has a seasonal cycle, so is not steady?

**page 11, line 6:** "the light extinction factor was calculated considering a background value"

**page 11, line 26:** well-lighted -> well-lid

**figure 6:** caption: what is LSn?

**page 11, line 33-34:** ". . . to solve . . . "

**page 11, line 36:** how does the scaling with numbers of tracers emerge from the figure? What experiments were done in regard?

**page 11, line 40:** 2400s = 40' not 45'? What does LSn mean, I suppose number of time steps per long step, but should be specified.

**figure 7:** Eta?

**page 12, line 36:** connected to the Ionian Sea rather than the Eastern Med.

**page 13, line 12:** daily fresh water flow rate

**page 13, line 35:** is the bulk background extinction coefficient adequate considering the amount of gelbstoff in the Northern Adriatic?

**page 14, lines7-9:** The large scale oscillations indicate instability rather than inaccuracy which may have increased earlier, considering that the dominant time scales will be different with respect to the gyre configuration. Or have you actually assessed the inaccuracy?

**page 13, line 39:** No benthic closure as described previously?

**page 14 line 33:** much lower mixed layer depth in winter

**figure 10:** Mention that phosphate is in contours.

**page 14, line 39:** "superimposed longitudinal gradient of the background light extinction": this is in contrast to the configuration description in line 35 of page 13 which mentions a constant background extinction.

**page 15, line 1:** consistent

**page 15, line 16-17:** drop "into the Mediterranean Sea"

**page 15/16, line 39-2:** A string matching mechanism using variable metadata would be more transparent.

**page 16, line 4-5:** Efficiency in spatial domain decomposition parallelisation is also considered in other coupling interfaces including the ones cited, so not unique to this interface. It may be more efficient, but that statement requires evidence.

---

## Referee Comment (RC2) · C. Lemmen (Referee) · 28 Dec 2016

In their manuscript, the authors present a new modular coupling scheme between the MIT hydrodynamic and the BFM biogeochemical model. They explore the numerical efficiency of this new coupled model and assess the trade-off between coupling time steps and simulation speed. The coupled model is successfully tested in an idealised and a realistic setup.

**General comments**

The manuscript's scientific significance is good. One could state that this is only yet

another example of a coupled system borne out of existing submodels. More than just combining the two existing submodels, however, the authors create an added value with their detailed description of the coupling process and with their conceptually modular approach to model coupling that can be (and should be) reused in the wider community for subsequent coupling attempts.

The scientific quality of the manuscript is very good. Methods are explained in detail, and conclusions are based solely on the material presented. I should be fairly straightforward to reproduce the work independently; some information on upgrading the GCHEM is missing.

The language used is clear and concise; the manuscript is well structured, all tables and figures are helpful. Unfortunately, the technical quality of the figures is not reaching the standard of the written material. All figures should improved or redesigned. At the current state, they are not acceptable for publication. Neither is the mathematical typesetting which also needs major improvements.

**Specific comments**

**Abstract** Within the abstract, the main properties of the BFM should be added (e.g, that it is a NPZD type model). The sentence "efficient scheme that manages communication and memory sharing" needs clarification, as it is not clear what "efficient" refers to (memory, time, ...). Also, tell the reader what the "expected theoretical" and the "observed" behaviour is.

**p1 l33f** There is much information on the feedback of BGC on hydrodynamics, but this feedback is not realized in the model presented. It is, however used to motivate the coupling of hydrodynamic and BGC models. Please disentangle.

**p2 l4** I don't agree with your wording "because of". The improvements are not causally

related to better computer resources; rather these increased resources have enabled the inclusion of more processes.

**p2 l6** Please provide a reference for increased "flexibility and modularity" of applications. This is neither (again) caused by increased resources, nor do I see much evidence for increased flexibility and modularity yet. What is your exact definition of modularity (and flexibility?)

**p2 l15f** I disagree with the statement "hydrodynamic codes are far more complex than BGC codes", and I disagree with the word "because". I think the reason is historic, because we started numerical simulations with hydrodynamic models and added more processes (among them BGC) onto the hydrodynamic model. Once physical oceanographers claimed their stakes in handling these hydrodynamic models with BGC "appendices" it became difficult for other disciplines to establish an alternate working mode, like building a BGC model where the hydrodynamics is the "simpler" appendix.

Moreover, the seemingly complicated (not complex!) mathematical description and notation used by physical oceanographers (see Eqs. 1–9 in this manuscript) might have excluded researchers from other fields to drive development of coupled hydrodynamic models. And while the maths involved in solving Eqs. 1–9 is certainly complicated, the many more interactions in biological systems make the ecology and biogeochemistry the more complex part of a coupled model system.

Talking about code complexity, much of the code base of physical ocean models is concerned with input and output, with infrastructure to define the model domain and its boundaries, and not the hydrodynamic core. BGC can be slimmer when they are coupled to such ocean models that provide all the infrastructure. And, of course, a major process (tracer advection diffusion) is shared by both BGC quantities and physical quantities and contributes to code complexity of either one (or both) the BGC and the physical model.

Interactive
comment

**p2 l16f** "is preferable" is a valuing statement that should be avoided. Also, the reason given "it facilitates upgrades" is not substantiated. In contrast, some might argue that a centralized/monolithic system is better to handle w.r.t. upgrading. Please elaborate (here or at a different suitable place)

**p2 l25ff** Name those existing couplings with MITgcm specifically, name the "specific high-complexity model". Why are those coupled models not "state-of-the-art"; if it is the lack of multi-nutrient/multi-species support, then make a causal statement.

**p2 l30** You give no motivation why us do not use one of the existing couplings of BFM with another hydrodynamic model. Please elaborate on your motivation to do yet another coupling to another ocean model. Also, I would like to see a discussion why you did not consider to include BFM in the FABM framework, as this would give you instant modular coupling to a multitude of ocean models that already implement FABM.

**p2 l33** "coupling tools were not used because ... want to preserve performance". Now that is exactly the raison d'etre for a coupling tool like ESMF, which has been proven to preserve performance and have a very low overhead. A more in-depth discussion for not choosing ESMF (or similar) is required at this point. And, of course, a better substantiation for your conclusion to disregard existing coupling tools.

**p2 l38** How do numerical accuracy and good performance lead to flexibility? I dont' understand your "Therefore". Throughout the manuscript, please make sure you argue both your points "performance" (which is well substantiated) and "flexiblity" (less so) consistently, or drop flexibility as a goal of your coupling if not substantiated better.

**p3 l6** sometimes, BGC models do *not* solve equations but provide tendencies only that are solved by the hydrodynamic model, as well.

**p3 l10** I would agree to ignore effects of BGC on hydrodynamics, but you use exactly this argument to motivate coupling in your introduction. Please resolve this conflict within your manuscript.

**p3 l15ff** The typesetting of the equations is poor and hinders understanding. It is unacceptable for publication, please use professional math typesetting. Major issues are font sizes and spacing within the equations, it is hard to see subscripts, implied multiplications appear in subscript, bold face is not clearly distinguishable ...

**p3 l27** Subscripts h and v (horizontal, vertical) not explained

**p3 l28** Acronym RHS = right hand side not explained

**p3 l29** There is not plain F, but subscripted F

**p4 l1** see above for typesetting Eqs. Here, especially, the problem with subscripts is apparent. I suggest they should be upright roman if words (like bio, BFM) and space to exclude misunderstanding as B * F * M implicitly.

**p4 l8** "can be handled". Who decides to handle Eq. 8? This is a serious modular coupling issue that deserves more discussion. What is your solution, specifically and why?

**p4 l22** All code and filenames should be in a typewriter font. (i.e starting from "use") and also "data.pkg"

**p4 l23** Check exact versioning and typesetting for model version of MITgcm

**p4 l24** Unclear typesetting, why all caps TRACERS? please set PTRACERS in typewriter font.

[Figure]

**p5 l1ff** Typesetting. You may consider to introduce a specific typesetting scheme for your code parts, but you need to explain it beforehand. This applies to the entire manuscript.

**p5 l1ff** Consider to put this information in a table.

**p6 l4** Where are these alterations "upgrades" to GCHEM documented? Where are they available (as a patch?, in the MITgcm distro?)

**p6 l36** The star "*" is not a mathematical symbol

**p6 l18** Do not extend the "≈" sign over the subscript

**p8 l11,19** inconsistent subscript "s" to PAR and many more equation typesetting problems

**p9 l35** Under what license is BFMcoupler distributed (MIT)? Also name the license for BFM (GPL) at the appropriate place earlier in the text. How do these licenses play together. Did you include any code parts from BFM into the coupler and how did you deal with changing the license?

**p9 l35** Who ensures long-term availability of the code if hosted on github? Could the currently used version (which git sha?) be archived and provided as SOM?

**p11 l40** Dt_trc is not explained, please check all your acronyms and subscripts for existing explanations. Consider a table of symbols.

**p12 l6f** Please motivate your choice of the 0–200 m depth range for integration of Chl. Maybe provide some infomation on the typical depth distribution of Chl in this region. Also, use correct typography (en-dash and not hyphen between number and unit).

**p12 l22** Provide evidence that the floating point rounding led to this error. How is the mass budget affected by different choices of the LS option?

**p13 l38** pCO2 not explained, also bad typography.

**p14 l5** for how many cores did you come up with a simulation time of 65 hrs per year?

**p15 l29** I don't agree with the term "framework". You describe a coupling and a coupling strategy, but certainly not a framework.

**p16 l3ff** This paragraph is entirely unclear to me. Both FABM and MESSy are capable of domain decomposition and have been shown to scale very well. Please better substantiate or argue your reasons not to use these somehow established coupling frameworks.

**p17 l10** I would recommend to add a section and paragraph "Code availability" here with the web resources, git/archive information, version numbers, and licenses. Move the content from appendix A4 here to this main section.

**Figure 1** Spelling of Omnivorous, Oxidation. From the figure, it is unclear what the superscript numbers (1) etc. denote, please explain. If (as I believe) this is a mere index, then the notation is rather unusual and confusing. Could you find a better notation or make transparent why you use this one? Why do you not use the subscript $i$ for your inorganic nutrients but spell them out, this is inconsistent.

**Figure 2** There is a need to synchronize all symbols with those used in the main text, also refer to the (suggested) table of symbols, if available. Some simplifications in terminology could help, why E-P-R, if EPR is sufficient—or call it moisture balance; why the sum of $Q_x$ where $Q_\theta$ is sufficient. Help the reader distinguish between C for carbon and C for concentration, especially as subscript; make sure the notation aligns with figure 1.

**Figure 3** Align typesetting with (revised) typesetting in-text. Clean up, e.g., why pTracer and PTRACER in the same figure? Please explain, how you numerically handle the equation pTracer=pTraders + dt*gchemTendency with regard to CFL and possible under/overflows.

**Figure 4** Thanks for choosing an accessible colormap! Avoid redundancy in graph (of geographic location), do not let axis labels overlap. Provide more context in the caption to which experiment (i.e. idealized closed basin) this graph belongs. Move unit from caption to colorbar; clarify whether this is magnitude of 3D speed or magnitude of speed projected on z-plane.

**Figure 5** Remove negative signs wherever possible. Depth is usually a positive number increasing from the surface do the bottom, adhere to this more common convention. Do not denote the contour with a minus sign: concentrations and NPP cannot be negative (that's the first impression I get...). Give color-coded and contoured quantity at the same place in the figure, like "color: T, contour: MLD". Make sure you explain all acronyms, sometimes a word might work instead of the symbol, e.g. "Temperature" instead of T. Be consistent with units and exponents. Move unit of color-coded quantity to colorbar. rotate the "m" unit on y-axis as it falsely gives the impression of an "E" at first glance; add "Depth" to y-axis to help the reader. Avoid overlap of numbers between graphs. Spell Hovmöller with a "v". Give context on experiment as in figure 4. Make sure font sizes are similar to Figure 4, 6 (and all other figures).

**Figure 6** Again, use consistent exponents, explain all acronyms. Add more meaning to the very technical caption. Pay attention to detail like the mismatch between the statement "root mean square error" and RMSD (unexplained, but I suppose "deviation"); add your choice of acceptable RMSD threshold to graph. Consider using stacked bars instead of area plot for left axis. Improve looks of figure.

**Figure 7** Explain "Eta" (or use greek symbol as suggested for text, and add semantics for sea surface height), tautology "alkalinity concentration", use consistent name "alkalinity" or "total alkalinity". Align style of figure to other figures in your manuscript. Replace "1 layer" with "surface layer". Improve overall look of figure (ticks, legend, line width/styles, labels, title ,....).

**Figure 8** Improve overall figure quality. Choose more contrast for depth colormap (consider established terrain/ocean floor colormaps); larger font sizes throughout. Unit of geographic coordinates is degree N or E. Add padding between text and figure margins. Make all named features (river entry points, site location, Otranto strait) more visible. Possibly combine with figure 9 showing 3 panels alongside? The line showing Otranto strait is not identical to the extent shown in figure 11.

**Figure 9** Avoid redundancy in y-axis (Latitude); be consistent with unit (see figure 8), it is not helpful to show 10 times "N" along the axis. S not explained, again different notation for chlorophyll (CHLa, in other places Chl). Add information on countrou quantity in the plot (near the colorbar). Panel a) surface current magnitude needs a scale bar. Panel b) I prefer Chl to be shown on a log scale. Blue contour unfortunately collides with blue color in Ionian Basin, try different color for this contour.

**Figure 10** Negative depth axes, log scale, colliding lines and text, font sizes, misleading y-axis label... please pay attention to these details. Suggestion: add "Adriatic Pit" and "Ionian deep" to a) and b) to improve intuitive access to figure. Visually separate years and spell out the full year 2008 etc. under the x-axis. Add "contour" in caption,

**Figure 11** Negative depth axes, colliding lines and text and colliding texts, missing axis labels, undefined acronyms ... Avoid redundant x-axis (longitude) information, add units to colorbar, add contour info within figure. Consider different scaling of colored quantities, e.g. log or square root (with workaround for negative values)

to show more detail in the small numbers. Added semantics like "Vertical section" to caption.

**Technical corrections**

**p2 l7** "is" should be "has become"

**p2 l40** "following" should be "subsequent"

**p3 l8** delete "very"

**p4 l22** composed of

**p4 l31f** use a proper em-dash and not a hyphen

**p5 l11** "custom" instead of "customary"

**p11 l25** well-lit, not "well-lighted"

**p12 l17** "eta" should $\eta$?

**p13 l35ff** use consistent exponent, e.g, either m$^{-1}$ or $\frac{1}{m}$ or $1/m$, but only use one of those styles in the text. I prefer the first with negative exponent. Check entire manuscript.

**p15 l1** "consistent" not "consistently"

---

## Author Comment (AC1) · 15 Feb 2017

**Letter to reviewers**

Dear Editor,

please find below our answers to the reviewers' comments.
In the new version of the manuscript we carefully revised all the figures and the typesetting of the equations, variables and pieces of code (as recommended by Rev#2). The abstract, the introduction and the discussion have been modified according to the suggestions of Rev#2. Finally, we also provided the missing technical information and we amended the scientific description of some topics, which was rather imprecise (as pointed out by Rev#1).

In the response letter, the comments of the reviewers are in *ITALIC SMALL CAPITAL* whereas author's responses are in upright font. A revised manuscript has also been uploaded. We provided a marked-up version of the manuscript showing the changes we made (we used the track changes in Word).
We hope that the revision will meet the high standards of Geoscientific Model Development publications.

Best regards,

Gianpiero Cossarini and co-authors

*REVIEWER #1*

*1.1      GENERAL REMARKS*
*THIS MANUSCRIPT PRESENTS A SOFTWARE INTERFACE THAT LINKS TWO WELL ESTABLISHED MODELS WITHIN THE MARINE SCIENCE COMMUNITY FOR OCEAN CIRCULATION AND MARINE BIOGEOCHEM-ISTRY, PROVIDING A NEW VALUABLE TOOL FOR A VAST RANGE OF APPLICATIONS INDICATED IN THE WORK. I'D LIKE TO COMPLIMENT THE AUTHORS FOR THEIR EFFORT IN COVERING THE WIDE RANGE ARGUMENTS INVOLVED IN DESCRIBING A COUPLED HYDRODYNAMIC BIOGEOCHEMICAL SYSTEM COMPREHENSIV- LEY IN A REASONABLE AMOUNT OF SPACE WITH AN ADEQUATE LEVEL OF DETAIL WITHOUT ANY MAJOR OMISSION. THE WORK COVERS ALL RELEVANT SCIENTIFIC AND TECHNICAL ASPECTS AND IS THEREFORE CERTAINLY ELIGIBLE FOR PUBLICATION IN GEOSCIENTIFIC MODEL DEVELOPMENT AFTER THAT SOME MINOR ISSUES HAVE BEEN ADDRESSED WHICH CONCERN SOME MISSING TECHNICAL INFORMATION AND INPRECSIONS IN THE SCIENTIFIC DESCRIPTION, THAT ARE GIVEN BELOW.*
We thank the reviewer for the positive comment. We improved the quality of the manuscript incorporating all the missing information and the corrections suggested by the reviewer.

*1.2      TECHNICAL OMISSIONS*
*IN ORDER TO BENEFIT USERS OF THIS SOFTWARE INTERFACE, I BELIEVE THE ADDITION OF THE FOLLOWING INFORMATIONS WOULD BE CRUCIAL:THE VERSION OF THE MITGCM USED IN THIS WORK HAS BEEN SPECIFIED, BUT AN INDICATION OF WHERE TO RETRIEVE THE MODEL CODE FROM WOULD BE HELPFUL. OF COURSE, NOT BEING THE DEVELOPERS THEMSELVES, THE AUTHORS WILL HAVE NO CONTROL OF THE FUTURE ACCESSIBILITY TO THE POINT OF RETRIEVAL, BUT AT LEAST AN INDICATION OF WHERE OR HOW TO OBTAIN THE CODE AT PRESENT IS REQUIRED.*
That information was missing: we specified the MITgcm website and the two possible ways to obtain the code in P6L7-12 The link to download the TAR file (http://mitgcm.org/download/) and the link to the CVS pserver (http://mitgcm.org/public/using_cvs.html) can be accessed very easily from the MITgcm home, so we don't think it is worth specifying them in the text. Similarly for the terms TAR and CVS, which are quite common in the modeling community and which we don't think that should be explained.

*WHAT IS THE PRECISE VERSION OF THE BFM CODE, THAT WAS USED? I GATHER THE MODEL IS DISTRIBUTED VIA GIT, SO A COMMIT HASH AND THE LINK TO THE REPOSITORY WOULD SUFFICE.*
The current coupling uses the BFM version v2, which was used in several papers  describing the biogeochemistry of the Mediterranean Sea (Lazzari et al.,2012, 2016; Teruzzi et al., 2013; Cossarini et al., 2015; Melaku Canu et al., 2015).
As we explain in the text, the BFM can be downloaded by registering and requesting the code through the BFM website (http://bfm-community.eu).
The text has been modified as follows: "For this application, we adopted the configuration version v2 (Lazzari et al. 2012, 2016; Teruzzi et al., 2013, Melaku Canu et al., 2015, Cossarini et al., 2015b), which can be downloaded upon request from the BFM-consortium.eu website " at P7L31-32.

*PROGRAMMING LANGUAGES AND VERSION OF ALL PARTS SHOULD BE SPECIFIED. TO MY UNDERSTANDING ALL THREE ARE CODED IN FORTRAN, BUT ONLY FOR THE BFM THIS IS CLEARLY STATED INCLUDING THE FORTRAN VERSION.*
We specified that MITgcm is a modular Fortran77 code in P6L9, and that BFM is Fortran90 code in P7L14.

*AS I ASSUME DEVELPMENT OF THE COUPLE WILL CONTINUE, ALSO A HASH FOR THE COUPLER LIBRARY IS NEEDED.*
Yes, this is the first release of the BFMCOUPLER package and its development will continue. It has been named v1.0. The version is added BFMCOUPLER in the title of the manuscript, in the Appendix at P23L6 and L11 and P26L11 and in the headers of all BFMCOUPLER files available in the GitHub repository.

*SPECIFICATIONS OF INPUT/OUTPUT FORMATS ARE MISSING.*
Input/output of the coupled model is based on the native MITgcm I/O package (MDSIO), which is a package that contains a group of Fortran routines for reading and writing direct-access binary files.
A sentence has been added in the description of the BFMCOUPLER package in the Appendix A at P24L21-22: "Input/output directives are based on the native MITgcm I/O package (MDSIO), a set of Fortran routines for reading and writing direct-access binary files".

**1.3    DETAILED COMMENTS**

*PAGE 2, LINE 12: CHECK AUTHOR NAMES AND PUBLICATION YEAR.*
Butenschön et al., 2016 has been corrected

*PAGE 2, LINE 15: I DON'T THINK IT'S A MATTER OF COMPLEXITY, BUT A STRUCTURAL ARGUMENT THAT DICTATES THE MODEL HIERARCHY BETWEEN PHYSICAL DRIVER AND BIOGEOCHEMICAL MODEL: IN ALL CITED CASES AND ALL OTHER COUPLINGS I AM AWARE OF, THE PHYSCIAL MODEL PRO- VIDES THE OVERARCHING GEOMETRICAL STRUCTURE OF THE MODEL (AND CAN RUN ON ITS OWN), WHILE THE BIOGEOCHEMICAL MODEL TYPICALLY JUST COMPUTES BIOGEOCHEMICAL RATES ON A PER PIXEL BASE AND AT THE BARE MINIMUM REQUIRES A 0D DRIVER TO RUN ANY SIMULA- TION. IN FACT, I DON'T THINK THAT THE STATEMENT THAT THE PHYSICAL MODELS ARE FAR MORE COMPLEX HOLDS GENERALLY.*
The reviewer is right: the sentence is rather misleading. Please see also Rev#2's comments about this issue (P2L15F of Rev#2). The sentence has been corrected as follows: " … , in general, hydrodynamic models have been already developed to solve the partial differential equation of tracers and provide the coding infrastructure to handle the spatial-temporal properties of the simulations (i.e. bathymetry, boundaries, computational domain discretization) " at P2L25-27.

*PAGE 2, LINE 18: INSTEAD OF "INPUT/OUTPUT DIRECTIVES" I'D PREFER INTERFACE SPECIFICATIONS OR IF YOU WANT MORE USE THE TECHNICAL TERM APPLICATION PROGRAMMING INTERFACES (APIS) IN ORDER TO NOT CONFUSE WITH ACTUAL MODEL IN- AND OUTPUT DATA.*
Thank you for the suggestion: the term "application programming interfaces (APIs)" has been used instead of "input/output directives" at P2L29, P8L8, P23L10.

*PAGE 2, LINE 32-34: THE CHOICE OF THE AUTHORS IN ITSELF DOESN'T REQUIRE JUSTIFICATION, BUT IS THERE ANY EVIDENCE THAT PERFORMANCE WOULD SUFFER FROM USING THESE TOOLS? I WOULD HAVE THOUGHT THAT DEPENDS (TO SOME DEGREE) ON HOW WELL THEY ARE WRITTEN?*
This part of the introduction has been significantly revised considering also the comments of Rev#2.
In the revised text, we explain the motivations for our online coupling (i.e., capability to drive the biogeochemistry at the same frequency of the hydrodynamic processes, to avoid the use of large files in which to save hydrodynamic variables at high frequency, to ensure the use of consistent differential operators - advection and diffusion - for hydrodynamics and biogeochemistry and to describe possible feedbacks from biogeochemistry to hydrodynamics). We removed the conjectures regarding other couplers.

*PAGE 2, LINES 34-38: WORTH MENTIONING HERE IF IT DEALS WITH ONLINE, OFFLINE COUPLING OR BOTH.*
"Online" is added at P3L16.

*PAGE 4, LINE 36: ANY EXPLICIT TIME INTEGRATION METHOD IS FORWARD-IN-TIME, THAT INCLUDES ADAMS-BASHFORTH SCHEMES. WHICH ONE IS USED HERE, EULER FORWARD?*
Yes. We specified Euler "forward-in-time" in P6L22 and L28, and also "backward" in P6L21.

*EQUATION 10: THE MEANING OF THE HERETO UNUSED VARIABLES NEEDS TO BE SPECIFIED. SPECIFICALLY THE MEANING OF THE TWO DIFFUSION TERMS SHOULD BE MADE CLEAR. ALSO, AS THESE EQUATIONS ARE STILL GENERIC AT THIS POINT, IT MAY BE WORTH USING A DIFFERENT SUB-SCRIPT FOR THE BIOGEOCHEMICAL SOURCES AND SINKS.*
We agree with the reviewer. Equation (10) has been corrected by substituting $\mathbf{R}_{BFM}$ with $\mathbf{R}_{bio}$, which is first introduced in equation (7) and named at P5L5. Further, the two diffusion terms of equation (10) have been explained in the sentence: "The first three …, where $K_H$ and $K_V$ are the horizontal and vertical diffusivities, respectively, which are considered separately because they have different spatial scales." at P5L15-16.

*PAGE 4, LINES 9-15: MIGHT BE WORTH ALSO TO REFER ALREADY TO THE POSSIBILITY OF SURFACE/BOTTOM BOUNDARY CONDITIONS FOR THE BIOGEOCHEMISTRY INTRODUCED IN THE BFMCOUPLER SECTION LATER.*
We agree to introduce here the surface and bottom forcing and we modified the sentence as follows: "The other components, such as Eq. (8), the biogeochemical tracers forcing terms ($\mathbf{Q_c}$ e.g. surface and bottom boundary conditions) and the sinking terms, can be handled by both the hydrodynamic and biogeochemical models" at P5L20-21. However, since this is still a general description, we would prefer to postpone to the section dedicated to the coupler (section 2.4) the description of the components and processes that are handled directly by the coupler.

*PAGE 6, LINES 16-17: THE INFORMATION FLOW BETWEEN BFM, MITGCM AND THE COUPLER IN BE- TWEEN IS NOT ONE-WAY AS CLEARLY INDICATED BY FIGURE 2 AND THE FOLLOWING TEXT. IT IS TRUE THAT THE OCEAN PHYSICS IN THE DESCRIBED SETTING REMAIN UNAFFECTED BY THE BIOGEOCHEMISTRY, BUT THAT DOESN'T MEAN THAT THE INFORMATION FLOW IS ONE-WAY, AS THE TRANSPORT MODEL FOR PASSIVE TRACERS SITS IN THE MITGCM CODE AND REQUIRES THE BIOGEOCHEMICAL SOURCES AND SINKS FOR INTEGRATION (AND BIOGEOCHEMICAL VERTICAL MOVEMENT).*
We agree, "one way" has been removed and the sentence has been rephrased as follows: "As an interface, the BFMCOUPLER manages the transfer of information that is required by the BFM from both the hydrodynamic and transport sub-models of the MITgcm, and provides the integration solver (a MITgcm package) with the biogeochemical surface and bottom forcing and the sink/sources terms originated from the BFM ( $gTracer_{bio}$ )." at P8L25-26.

*PAGE 6, LINE 33: "A SOURCE SPLITTING SCHEME IS ADOPTED": THE INSERTION OF THE BIOGEOCHEMICAL RATES INTO THE TRANSPORT SOLVER CONSTITUTES A SOURCE SPLITTING ONLY OF THE INTEGRATION SCHEME APLLIED EFFECTIVELY OMITS INFORMATION OR INTERMEDIATE STEPS THAT ARE REQUIRED BY THE COUPLER. IN PARTICULAR, IF THE INTEGRATION SCHEME IS A SIMPLE EULER FORWARD SCHEME, THE INSERTION OF THE BIOGEOCHEMICAL RATES DOES NOT INVOLVE ANY FORM OF SPLITTING, BUT CONSTITUTES A DIRECT INTEGRATION OF THE TERMS IN EQUATION 10.*
We agree with the reviewer, the use of the term "source splitting scheme" is misleading. We intended a source splitting scheme with synchronous time steps, that was then compared to the operator splitting scheme with different time steps. However, we agree that the term source splitting, as described in the original paper, is misleading. Therefore we decided to substitute "source splitting" with "direct integration scheme" throughtout the text.

*PAGE 7, LINES 23-24: THE OPTION OF APPLYING DIFFERENT TIME STEPS TO THE TWO MODES IS NOT UNIQUE TO THE PROCESS SPLITTING, BUT CAN BE EASILY ADOPTED IN THE SOURCE SPLITING METHOD BY UPDATING THE RATES OF THE SLOWER PROCESS ONLY ON INTERMEDIATE TIME STEPS. (E.G. BLOM, J.G., VERWER, J.G., 2000. A COMPARISON OF INTEGRATION METH- ODS FOR ATMOSPHERIC TRANSPORT-CHEMISTRY PROBLEMS. JOURNAL OF COMPUTATIONAL AND APPLIED MATHEMATICS 126, 381–396. DOI:10.1016/S0377-0427(99)00366-0)*
As explained at the previous point, the term "source splitting" has been dropped. We agree with reviewer's comment and, in order to be more clear, we referred to "direct integration" instead of "source splitting".

*PAGE 7, LINE 26-27: "TO INCREASE THE COMPUTATIONAL PERFORMANCE OF THE ENTIRE CODE" IN WHICH WAY?*
We inteded the overall time required to perform a coupled simulation. The sentence has been changed as follows: "A third option is a operator splitting algorithm, which involves the MITgcm package LONGSTEP (Adcroft et al., 2016) and adopts different time steps for the hydrodynamic and transport-biogeochemical components, thus increasing the computational performance of a coupled simulation" at P10L12-14.

*PAGE 8, LINE 15 FF, EQ 14: WHAT WOULD ACTUALLY BE REQUIRED IS THE AVERAGE LIGHT IN EACH CELL, WHICH GIVEN THE EXPONENTIAL DISTRIBUTION IS NOT THE LIGHT IN THE CELL CENTRE. USING THE SAME FORMULA, IT IS STRAIGHT FORWARD TO COMPUTE THE INTEGRAL OF LIGHT BETWEEN THE UPPER AND LOWER CELL FACE AND DIVIDE BY THE CELL THICKNESS IN ORDER TO ARRIVE AT THE CORRECT NUMBER.*
The current version of the BFM model (version v2, which is used for Mediterranean Sea simulations) was calibrated (i.e., parameters of the phytoplankton growth formulation) to use the light at the center of the cell (Lazzari et al., 2012, 2016). As we described in the revised section 2.3 (P7L29-31) we specify that we use the v2 configuration of the BFM model, which is fully described in the aforementioned papers. The light formulation is consistent with the BFM model implementation, however, we agree that it is easy to accomplish the implementation of alternative light locations (i.e., average light in the cell or light at the top of the cell). Finally, considering the fine discretization of the vertical dimension (cell thickness lower than 5 m down to 45 m and lower than 10 m down to 120 m), the differences between PAR at center of the cell and integral PAR of the cell are quite low (less than 6% down to 45 m and less than 11% down to 130 m).

*PAGE 8, EQ. 15: HOW IS THIS PDE SOLVED NUMERICALLY? PROVIDE THE SCHEME OR A REFERENCE TO A FULL DESCRITION.*
The sinking is solved numerically based on an Euler forward scheme. This is added to the text at P12L2-3.

*PAGE 9, LINE 5: WHAT HAPPENS TO SINKING PHYTOPLANKTON THAT HITS THE SEA FLOOR?*
The solution of the sinking equation (Eq. 15) has a boundary at the last cell of the water column. There is no sinking flux from the last cell to the sea bottom. However, in order to avoid misunderstandings, we have substituted the word "sink out" with "is exported out from" in the sentence that describes the burial process at P12L16. In the case of burial, it is assumed that a fraction of the detritus concentration (but not of the phytoplankton) of the last cell of the water column hits the sea floor and exits from water column.

*PAGE 9, LINE 28: ". . . WHICH ARE . . . "*
Done. P13L11

*PAGE 10, LINE 6: FURTHER DOWN IT APPEARS THAT AT LEAST THE SURFACE HEAT HAS A SEASONAL CYCLE, SO IS NOT STEADY?*
Yes, surface heat and mass fluxes have a seasonal cycle, the adjective "steady" is referred to wind only. The sentence has been made clearer as follows: "… forced by steady winds and a seasonal cycle of surface heat (downward long-wave and short-wave radiation) and mass (precipitation) fluxes" at P13L27
Further, in Appendix A at P26L9-10, the sentence that describes the idealized case, which can be downloaded along with the code, has been corrected as well.

*PAGE 11, LINE 6: "THE LIGHT EXTINCTION FACTOR WAS CALCULATED CONSIDERING A BACKGROUND VALUE".*
Done: "s" has been removed. Further, the value of the background extinction factor has been added as follows: "… considering a background value ($K_{ext}$=0.035 m$^{-1}$) …" at line P15L3.

*PAGE 11, LINE 26: WELL-LIGHTED -> WELL-LID*
Done.

*FIGURE 6: CAPTION: WHAT IS LSN?*

$LS_n$ is the number of ocean dynamics time steps performed within a single ptracer step. The explanation of $LS_n$ has been introduced in the text at P10L14 and reported in the caption of Figure 6 for sake of clarity.

*PAGE 11, LINE 33-34: "... TO SOLVE..."*
Done. P15L31

*PAGE 11, LINE 36: HOW DOES THE SCALING WITH NUMBERS OF TRACERS EMERGE FROM THE FIGURE? WHAT EXPERIMENTS WERE DONE IN REGARD?*
We performed four tests using the PTRACER package without the biogeochemical component and varying the number of passive tracers (10, 20, 40 and 51). We decided not to show these results, which prove that the computation cost increases almost linearly with the number of the passive tracers, because we thought they were quite "trivial" and predictable. Further, an indirect proof of this result is also shown in Figure 6. The solution of temperature and salinity equations (i.e., the transport of two tracers) requires almost 390 s, whereas the solution of transport of the 51 biological tracers requires about 9000 s. So, the transport of each tracer is solved in almost 190-200 s, and the computational cost increases roughly with the number of the tracers.
We have added the following "(e.g., Tracers$_{trsp}$ is almost 25 times larger than the time used to solve for temperature and salinity; Fig. 6)" in the text at P16L2-3.

*PAGE 11, LINE 40: 2400s = 40' NOT 45'? WHAT DOES LSN MEAN, I SUPPOSE NUMBER OF TIME STEPS PER LONG STEP, BUT SHOULD BE SPECIFIED.*
Done: $LS_n$ is now first introduced at P10L15; and the sentence has been rewritten as follows "With a $LS_n$ sets to 8 (a time step for tracers, $\Delta t_{trc}$, equals to 2400 s), the …" at P16L7.

*FIGURE 7: ETA?*
Eta has been substituted with Sea Surface Height (SSH).

*PAGE 12, LINE 36: CONNECTED TO THE IONIAN SEA RATHER THAN THE EASTERN MED.*
Done. Ionian Sea instead of Eastern Mediterranean Sea at P17L12.

*PAGE 13, LINE 12: DAILY FRESH WATER FLOW RATE*
Done. "fresh water" has been added at P17L28.

*PAGE 13, LINE 35: IS THE BULK BACKGROUND EXTINCTION COEFFICIENT ADEQUATE CONSIDERING THE AMOUNT OF GELBSTOFF IN THE NORTHERN ADRIATIC?*
Background extinction coefficient has been set considering a longitudinal gradient, according to the results presented in Lazzari et al., 2012. The sentence, which erroneously reported a constant background extinction coefficient, has been changed as follows "The background water light extinction coefficient was set considering a longitudinal negative gradient according to Lazzari et al. (2012)" at P18L20.
In any case, we think that the study of the effect of riverine yellow-substances on the water transparency in the northern Adriatic Sea would require a dedicated investigation, which should simulate the dynamical evolution of a new state variable for the terrestrial detritus concentration (yellow substance). An alternative solution would be the use of satellite maps of Kd as background extinction coefficient. In both cases, the coupled model has all the elements in order to carry out such an investigation.

*PAGE 14, LINES 7-9: THE LARGE SCALE OSCILLATIONS INDICATE INSTABILITY RATHER THAN INACCURACY WHICH MAY HAVE INCREASED EARLIER, CONSIDERING THAT THE DOMINANT TIME SCALES WILL BE DIFFERENT WITH RESPECT TO THE GYRE CONFIGURATION. OR HAVE YOU ACTUALLY ASSESSED THE INACCURACY?*
The accuracy has been computed as the annual average of the root mean square of the differences of the weekly 3D fields between the reference solution ($LS_n = 1$) and the solution obtained when $LS_n$ equals 3, 6, 9 and 12. This has been explained in the caption of Table 2.

The sentence has been simplified, avoiding comments on results which are not shown: "Then, time steps higher than 30 minutes substantially decreased the accuracy, without further reducing the computational cost (Table 2)" at P19L1-2.

*PAGE 13, LINE 39: NO BENTHIC CLOSURE AS DESCRIBED PREVIOUSLY?*
Yes, the set up of the Adriatic-Ionian model doesn't have bottom forcing because very few information are available on bottom fluxes and their temporal and spatial variability in the study area. We think that the investigation of the bottom-water column interaction effects would deserve a dedicated investigation, since it can have important effects on the productivity in the shallow continental shelf areas. A comment about the importance of coupling the present model with a benthic sub-model is present in the discussion section. Thus, we mostly focused our presentation on the results of the pelagic area (e.g., Figure10: vertical structure of variables in open sea points; Figure 11: transport of carbon across the Otranto strait).

*PAGE 14 LINE 33: MUCH LOWER MIXED LAYER DEPTH IN WINTER FIGURE 10: MENTION THAT PHOSPHATE IS IN CONTOURS.*
Done; then, Figure 10 has been redrawn considering also the comments of Rev#2.

*PAGE 14, LINE 39: "SUPERIMPOSED LONGITUDINAL GRADIENT OF THE BACKGROUND LIGHT EXTINCTION": THIS IS IN CONTRAST TO THE CONFIGURATION DESCRIPTION IN LINE 35 OF PAGE 13 WHICH MENTIONS A CONSTANT BACKGROUND EXTINCITION.*
As described at the point *PAGE 13, LINE 35*, the background extinction coefficient has been set considering a longitudinal gradient, according to the results presented in Lazzari et al., 2012. The sentence, which erroneously reported a constant background extinction coefficient, has been changed as follows "The background water light extinction coefficient was set considering a longitudinal negative gradient according to Lazzari et al. (2012)" at P18L20.

*PAGE 15, LINE 1: CONSISTENT*
Done.

*PAGE 15, LINE 16-17: DROP "INTO THE MEDITERRANEAN SEA"*
We would prefer to leave the term "into the interior of the Mediterranean Sea", because the net transport of carbon through the Otranto strait from the Adriatic occurs at the deeper layers. Thus, we would like to convey the message that the carbon is then entrapped in the deep water masses of the Mediterranean Sea.

*PAGE 15/16, LINE 39-2: A STRING MATCHING MECHANISM USING VARIABLE METADATA WOULD BE MORE TRANSPARENT.*
To our understanding, this would necessitate the use of pointers that systematically link the MITgcm ptracers to the BFM variables for the formulations used in the `BFMCOUPLER`. This could result in an increase of the complexity of the coding. The actual programming effort to link BFM variables and MITgcm tracers is quite small. Indeed, the list and order of the BFM variables are described only once in the include files `BFMCOUPLER_VARS.h` and `BFM_var_list.h`; and in the input file `data.ptracer`.

*PAGE 16, LINE 4-5: EFFICIENCY IN SPATIAL DOMAIN DECOMPOSITION PARALLELISATION IS ALSO CONSIDERED IN OTHER COUPLING INTERFACES INCLUDING THE ONES CITED, SO NOT UNIQUE TO THIS INTERFACE. IT MAY BE MORE EFFICIENT, BUT THAT STATEMENT REQUIRES EVIDENCE.*
We agree with the reviewer, this sentence was not clear (see also Rev#2 on this issue). We wanted to state that the efficiency of a coupled code can be an issue. Our coupling (as others like FABM and MESSy) is capable to handle to this aspect.
The sentence has been rewritten as follows: "Despite the growth of computational resources, the efficiency of coupled codes can be still an issue because of the large size of the computational grids (Blom and Verwer, 2000). Domain decomposition and parallelization tools are available in several coupling environments (e.g., FABM, Bruggeman and Bolding, 2014; and MESSy, Jöckel et al., 2008). Likewise, our coupling scheme has been thought to fully exploit the parallelization efficiency of the MITgcm (Marshall et al., 1997), and no additional coding effort (in terms of parallelization) is required by the users" at P21L13-17.

*IN THEIR MANUSCRIPT, THE AUTHORS PRESENT A NEW MODULAR COUPLING SCHEME BETWEEN THE MIT HYDRODYNAMIC AND THE BFM BIOGEOCHEMICAL MODEL. THEY EXPLORE THE NUMERICAL EFFICIENCY OF THIS NEW COUPLED MODEL AND ASSESS THE TRADE-OFF BETWEEN COUPLING TIME STEPS AND SIMULATION SPEED. THE COUPLED MODEL IS SUCCESSFULLY TESTED IN AN IDEALISED AND A REALISTIC SETUP.*

*GENERAL COMMENTS*

*THE MANUSCRIPT'S SCIENTIFIC SIGNIFICANCE IS GOOD. ONE COULD STATE THAT THIS IS ONLY YET ANOTHER EXAMPLE OF A COUPLED SYSTEM BORNE OUT OF EXISTING SUBMODELS. MORE THAN JUST COMBINING THE TWO EXISTING SUBMODELS, HOWEVER, THE AUTHORS CREATE AN ADDED VALUE WITH THEIR DETAILED DESCRIPTION OF THE COUPLING PROCESS AND WITH THEIR CONCEPTUALLY MOD- ULAR APPROACH TO MODEL COUPLING THAT CAN BE (AND SHOULD BE) REUSED IN THE WIDER COM- MUNITY FOR SUBSEQUENT COUPLING ATTEMPTS.*

*THE SCIENTIFIC QUALITY OF THE MANUSCRIPT IS VERY GOOD. METHODS ARE EXPLAINED IN DE- TAIL, AND CONCLUSIONS ARE BASED SOLELY ON THE MATERIAL PRESENTED. I SHOULD BE FAIRLY STRAIGHTFORWARD TO REPRODUCE THE WORK INDEPENDENTLY; SOME INFORMATION ON UPGRADING THE GCHEM IS MISSING.*

*THE LANGUAGE USED IS CLEAR AND CONCISE; THE MANUSCRIPT IS WELL STRUCTURED, ALL TABLES AND FIGURES ARE HELPFUL. UNFORTUNATELY, THE TECHNICAL QUALITY OF THE FIGURES IS NOT REACHING THE STANDARD OF THE WRITTEN MATERIAL. ALL FIGURES SHOULD IMPROVED OR REDESIGNED. AT THE CURRENT STATE, THEY ARE NOT ACCEPTABLE FOR PUBLICATION. NEITHER IS THE MATHEMATICAL TYPESETTING WHICH ALSO NEEDS MAJOR IMPROVEMENTS.*

We thank Dr. Lemmen for the positive comments and the important issues he raised. We improved the quality of the manuscript revising all the figures and the typesetting of the equations, variables and pieces of code. Furthermore, we carefully revised the abstract, the introduction, the model descriptions and the discussion according to the proposed suggestions.

*SPECIFIC COMMENTS*

*ABSTRACT WITHIN THE ABSTRACT, THE MAIN PROPERTIES OF THE BFM SHOULD BE ADDED (E.G, THAT IT IS A NPZD TYPE MODEL). THE SENTENCE "EFFICIENT SCHEME THAT MANAGES COMMUNICATION AND MEMORY SHARING" NEEDS CLARIFICATION, AS IT IS NOT CLEAR WHAT "EFFICIENT" REFERS TO (MEMORY, TIME, ...). ALSO, TELL THE READER WHAT THE "EXPECTED THEORETICAL" AND THE "OBSERVED" BEHAVIOUR IS.*

Thank you for the remarks: we revised the abstract as follows. We specified that BFM is a model based on plankton functional types formulation. We better introduced the characteristics of the new coupler, describing that is online, open source and that has been developed in a way that preserves the sustainability of the programming effort to handle future evolutions in the two codes.

Finally, we specified that our model reproduces the alternation of surface bloom and deep chlorophyll formation driven by the seasonal cycle of winter vertical mixing and summer stratification in the mid latitude gyre test case. Then we reported that the main features and spatial patterns of the hydrodynamic and biogeochemical variables in the Mediterranean domain are consistent with the literature.

*P1 L33F THERE IS MUCH INFORMATION ON THE FEEDBACK OF BGC ON HYDRODYNAMICS, BUT THIS FEEDBACK IS NOT REALIZED IN THE MODEL PRESENTED. IT IS, HOWEVER USED TO MOTIVATE THE COUPLING OF HYDRODYNAMIC AND BGC MODELS. PLEASE DISENTANGLE.*

We thank Dr. Lemmen for his thorough comments. Considering also other comments regarding the coupling issues (*P1L33, P2L6, P2L30 AND P3L10*), we have significantly modified the introduction, amending several parts. In particular, we have introduced a new paragraph (new text at P3L8-15) including the rationales for an online coupling of hydrodynamics and biogeochemistry, and we explained our motivations for choosing to couple MITgcm and BFM.

In doing that, we moved the sentences regarding the influences (and feedbacks) of physics on biogeochemistry to the new paragraph. In paricular, we explained that the motivations for online coupling are several (i.e., forcing the biogeochemistry at the same frequency as the hydrodynamic processes, avoiding the use of large files where to save hydrodynamic variables at high frequency, ensuring the use of consistent differential operators - advection and diffusion - for hydrodynamics and biogeochemistry) and the capability of describing possible feedbacks from biogeochemistry to hydrodynamics is just a potential that the online coupling could provide.

*P2 L4 I DON'T AGREE WITH YOUR WORDING "BECAUSE OF". THE IMPROVEMENTS ARE NOT CAUSALLY RELATED TO BETTER COMPUTER RESOURCES; RATHER THESE INCREASED RESOURCES HAVE ENABLED THE INCLUSION OF MORE PROCESSES.*

"Because of" has been removed, and the sentence has been revised. The concept we would like to communicate is that the development of models has become a cooperative and multidisciplinary task, rather than an individual effort, since the code complexity is increased because of the use of new programming paradigms (i.e., parallel programming) and the biogeochemical model has become more complex due to the inclusion of new variables and processes.

The sentences have been rewritten as follows: "In recent decades, the increasing availability of significant computational resources has allowed substantial improvements in hydrodynamic and biogeochemical models in terms of both temporal and spatial resolution of the simulations, which required new specific programming and coding expertise (i.e., code optimization and parallel programming). In addition, biogeochemical model complexity has increased through the inclusion of new variables and processes (Robson, 2014), and model development has become a cooperative and multidisciplinary task rather than an individual effort." at P2L13-17.

*P2 L6 PLEASE PROVIDE A REFERENCE FOR INCREASED "FLEXIBILITY AND MODULARITY" OF APPLICATIONS. THIS IS NEITHER (AGAIN) CAUSED BY INCREASED RESOURCES, NOR DO I SEE MUCH EVIDENCE FOR INCREASED FLEXIBILITY AND MODULARITY YET. WHAT IS YOUR EXACT DEFINITION OF MODULARITY (AND FLEXIBILITY?)*

Please consider the previous point: the concepts have been revised, the words removed and the sentences have been rewritten (See new sentences at P2L13-17).

*P2 L15F I DISAGREE WITH THE STATEMENT "HYDRODYNAMIC CODES ARE FAR MORE COMPLEX THAN BGC CODES", AND I DISAGREE WITH THE WORD "BECAUSE". I THINK THE REASON IS HISTORIC, BECAUSE WE STARTED NUMERICAL SIMULATIONS WITH HYDRODYNAMIC MODELS AND ADDED MORE PROCESSES (AMONG THEM BGC) ONTO THE HYDRODYNAMIC MODEL. ONCE PHYSICAL OCEANOGRAPHERS CLAIMED THEIR STAKES IN HANDLING THESE HYDRODYNAMIC MODELS WITH BGC "APPENDICES" IT BECAME DIFFICULT FOR OTHER DISCIPLINES TO ESTABLISH AN ALTERNATE WORKING MODE, LIKE BUILDING A BGC MODEL WHERE THE HYDRODYNAMICS IS THE "SIMPLER" APPENDIX. MOREOVER, THE SEEMINGLY COMPLICATED (NOT COMPLEX!) MATHEMATICAL DESCRIPTION AND NOTATION USED BY PHYSICAL OCEANOGRAPHERS (SEE EQS. 1–9 IN THIS MANUSCRIPT) MIGHT HAVE EXCLUDED RESEARCHERS FROM OTHER FIELDS TO DRIVE DEVELOPMENT OF COU- PLED HYDRODYNAMIC MODELS. AND WHILE THE MATHS INVOLVED IN SOLVING EQS. 1–9 IS CERTAINLY COMPLICATED, THE MANY MORE INTERACTIONS IN BIOLOGICAL SYSTEMS MAKE THE ECOLOGY AND BIOGEOCHEMISTRY THE MORE COMPLEX PART OF A COUPLED MODEL SYSTEM.*

*TALKING ABOUT CODE COMPLEXITY, MUCH OF THE CODE BASE OF PHYSICAL OCEAN MODELS IS CONCERNED WITH INPUT AND OUTPUT, WITH INFRASTRUCTURE TO DEFINE THE MODEL DOMAIN AND ITS BOUNDARIES, AND NOT THE HYDRODYNAMIC CORE. BGC CAN BE SLIMMER WHEN THEY ARE COUPLED TO SUCH OCEAN MODELS THAT PROVIDE ALL THE INFRASTRUCTURE. AND, OF COURSE, A MAJOR PROCESS (TRACER ADVECTION DIFFUSION) IS SHARED BY BOTH BGC QUANTITIES AND PHYSICAL QUANTITIES AND CONTRIBUTES TO CODE COMPLEXITY OF EITHER ONE (OR BOTH) THE BGC AND THE PHYSICAL MODEL.*

We thank Dr. Lemmen for the clarification. Indeed, the sentence was written a bit "hastily"… We revised the sentence pointing out that the inclusion of the biogeochemical models into hydrodynamic ones (and not vice-versa) have occurred (to our knowledge) mainly because the hydrodynamic models had already been coded to solve the partial differential equation of tracers and had a coding infrastructure capable to handle the spatial-temporal properties of the simulations (i.e. bathymetry, boundaries, computational domain discretization).

The sentence has been revised as follows: " … because hydrodynamic codes have been already developed to solve the partial differential equation of tracers and provide the coding infrastructure to handle the spatial-temporal properties of the simulations (i.e. bathymetry, boundaries, computational domain discretization)" at P2L25-27.

*P2 L16F "IS PREFERABLE" IS A VALUING STATEMENT THAT SHOULD BE AVOIDED. ALSO, THE REASON GIVEN "IT FACILITATES UPGRADES" IS NOT SUBSTANTIATED. IN CONTRAST, SOME MIGHT ARGUE THAT A CENTRALIZED/MONOLITHIC SYSTEM IS BETTER TO HANDLE W.R.T. UPGRADING. PLEASE ELABORATE (HERE OR AT A DIFFERENT SUITABLE PLACE)*

We agree with the reviewer that we cannot argue which coupling philosophy is the best, because it depends on many factors (e.g., type of the models, size of scientific community working on them). Here we wanted to state that there can been different philosophies for coupling models: merging one model into the other, or develop a modular interface between the two. This part of the introduction has been rewritten at P2L22-29.

*P2 L25FF* NAME THOSE EXISTING COUPLINGS WITH MITGCM SPECIFICALLY, NAME THE "SPECIFIC HIGH-COMPLEXITY MODEL". WHY ARE THOSE COUPLED MODELS NOT "STATE-OF-THE-ART"; IF IT IS THE LACK OF MULTI-NUTRIENT/MULTI-SPECIES SUPPORT, THEN MAKE A CAUSAL STATEMENT.

"State-of-the-art" has been removed. The issue we would like to communicate is that there have already been experiences of coupling of the two models (MITgcm and BFM) with other models: they are suitable for being coupled, but they have never been coupled together. The sentences have been revised as follows: "The two models are widely used, as described in the next sections, and have been already coupled with several other models. For example, the MITgcm has already been coupled to low- (Parekh et al., 2005; Follows et al., 2006) or intermediate-complexity (Hauck et al., 2013, Cossarini et al., 2015a) biogeochemical models for a few specific applications and to a specific high-complexity model (Dutkiewicz et al., 2009) to explore the theoretical aspects of intraspecific competition in plankton communities. On the other side, the BFM has already been coupled to POM (Polimene et al., 2006), NEMO (Vichi and Masina, 2009; Epicocco et al., 2016) and to the offline OGSTM, an upgraded version of OPA (Lazzari, et al 2012). A direct coupling between MITgcm and BFM has not been implemented yet. Thus, we developed a dedicated online modular coupler linking them. The new coupler is open source and allows to exploit the high potentiality of the two models, to preserve the sustainability of the programming effort and to handle future evolution of the two codes" at P3L8-15.

*P2 L30* YOU GIVE NO MOTIVATION WHY US DO NOT USE ONE OF THE EXISTING COUPLINGS OF BFM WITH ANOTHER HYDRODYNAMIC MODEL. PLEASE ELABORATE ON YOUR MOTIVATION TO DO YET ANOTHER COUPLING TO ANOTHER OCEAN MODEL. ALSO, I WOULD LIKE TO SEE A DISCUSSION WHY YOU DID NOT CONSIDER TO INCLUDE BFM IN THE FABM FRAMEWORK, AS THIS WOULD GIVE YOU INSTANT MODULAR COUPLING TO A MULTITUDE OF OCEAN MODELS THAT ALREADY IMPLEMENT FABM.

We revised this part of the introduction (please consider also our answer to point *P1L33F*). As regards the existence of FABM, we would like to inform the reader that alternative options for coupling hydrodynamic and biogeochemical models exist. Our choice for an online coupling based on a specifically developed coupler rises from both practical and historical reasons: in fact, we had good experiences in working with both MITgcm (e.g. Querin et al., 2013; Sannino et al., 2015) and BFM (e.g. Lazzari et al., 2012, 2016; Cossarini et al., 2015). Considering that, to our knowledge, FABM is neither coupled with MITgcm nor with BFM, and that quite some time is needed to acquire full competences for using a model, our object is to describe and provide the reader with a coupling scheme linking the two models, rather than testing different couplings of hydrodynamic and biogeochemical models. We acknowledge that it would be interesting to explore the potentials of FABM and to compare the efficiency of different couplers, however, this is beyond the scope of the present work.

*P2 L33* "COUPLING TOOLS WERE NOT USED BECAUSE … WANT TO PRESERVE PERFORMANCE". NOW THAT IS EXACTLY THE RAISON D'ETRE FOR A COUPLING TOOL LIKE ESMF, WHICH HAS BEEN PROVEN TO PRESERVE PERFORMANCE AND HAVE A VERY LOW OVERHEAD. A MORE IN-DEPTH DISCUSSION FOR NOT CHOOSING ESMF (OR SIMILAR) IS REQUIRED AT THIS POINT. AND, OF COURSE, A BETTER SUBSTANTIATION FOR YOUR CONCLUSION TO DISREGARD EXISTING COUPLING TOOLS.

We revised this part of the introduction (Please consider our response to point *P1L33F*).

*P2 L38* HOW DO NUMERICAL ACCURACY AND GOOD PERFORMANCE LEAD TO FLEXIBILITY? I DONT' UNDERSTAND YOUR "THEREFORE". THROUGHOUT THE MANUSCRIPT, PLEASE MAKE SURE YOU ARGUE BOTH YOUR POINTS "PERFORMANCE" (WHICH IS WELL SUBSTANTIATED) AND "FLEXIBLITY" (LESS SO) CONSISTENTLY, OR DROP FLEXIBILITY AS A GOAL OF YOUR COUPLING IF NOT SUBSTANTIATED BETTER.

We used improperly the word "flexibility", and it has been removed. The intoduction has been substaintally revised (please, see comments to point *P1L33F*). Additionally, we have removed the word "Therefore". Here we would like to summarize that our results show that the coupled MITgcm-BFM model guarantees mass conservation of chemicals, has good computational performace and provides reliable results. Without making conjectures with regard to other couplers, we belive and show that our open source coupling package can be a promising tool for investigating marine biogeochemistry at different spatial and temporal scales.

*P3 L6* SOMETIMES, BGC MODELS DO \*NOT\* SOLVE EQUATIONS BUT PROVIDE TENDENCIES ONLY THAT ARE SOLVED BY THE HYDRODYNAMIC MODEL, AS WELL.

Solve has been substituted with the more generic term "describe" at P3L29.

*P3 L10 I WOULD AGREE TO IGNORE EFFECTS OF BGC ON HYDRODYNAMICS, BUT YOU USE EXACTLY THIS ARGUMENT TO MOTIVATE COUPLING IN YOUR INTRODUCTION. PLEASE RESOLVE THIS CON FLICT WITHIN YOUR MANUSCRIPT.*

We substantially revised the part of the introduction dealing with this point. Please refer to our comments on the point *P1L33F*.

*P3 L15FF THE TYPESETTING OF THE EQUATIONS IS POOR AND HINDERS UNDERSTANDING. IT IS UNACCEPTABLE FOR PUBLICATION, PLEASE USE PROFESSIONAL MATH TYPESETTING. MAJOR ISSUES ARE FONT SIZES AND SPACING WITHIN THE EQUATIONS, IT IS HARD TO SEE SUBSCRIPTS, IMPLIED MULTIPLICATIONS APPEAR IN SUBSCRIPT, BOLD FACE IS NOT CLEARLY DISTINGUISHABLE ...*

The typesetting of all equations has been complitely revised, using the appropriate Microsoft Equation objects (rather than the "Insert/Equation" tool).

We have introduced a new section 2.1 (Nomenclature and units) that explains the typesetting and convention used throughout the paper. Finally, as suggested, we added a new table (table B.1 in new appendix B) containing all the symbols and variables used throughout the text.

*P3 L27 SUBSCRIPTS H AND V (HORIZONTAL, VERTICAL) NOT EXPLAINED*

Done at P4L25. A new table (Appendix B) reports all symbols and variables used throughout the text.

*P3 L28 ACRONYM RHS = RIGHT HAND SIDE NOT EXPLAINED*

Done at P4L26.

*P3 L29 THERE IS NOT PLAIN F, BUT SUBSCRIPTED F*

Done, " $\mathbf{F}_H$ and $F_V$ " added at P5L2.

*P4 L1 SEE ABOVE FOR TYPESETTING EQS. HERE, ESPECIALLY, THE PROBLEM WITH SUBSCRIPTS IS APPARENT. I SUGGEST THEY SHOULD BE UPRIGHT ROMAN IF WORDS (LIKE BIO, BFM) AND SPACE TO EXCLUDE MISUNDERSTANDING AS B * F * M IMPLICITLY.*

The equation has been revised, all tysetting corrected and BFM substituted with $\mathbf{R}_{bio}$ according to the suggestion of reviewer#1.

*P4 L8 "CAN BE HANDLED". WHO DECIDES TO HANDLE EQ. 8? THIS IS A SERIOUS MODULAR COUPLING ISSUE THAT DESERVES MORE DISCUSSION. WHAT IS YOUR SOLUTION, SPECIFICALLY AND WHY?*

The coupling problem is described here in generic terms. The next section (2.4) of the manuscript will explain and discuss how the different terms are handled by the different pieces of code in our specific coupling. In order to communicate the generality of the topic, we have rephrased the sentence as follows "can be handled by either the hydrodynamic or the biogeochemical model, according to the specific processes and the features of the codes" at P5L20-21.

*P4 L22 ALL CODE AND FILENAMES SHOULD BE IN A TYPEWRITER FONT. (I.E STARTING FROM "USE") AND ALSO "DATA.PKG"*

Done. Further, we have introduced a new section 2.1 (Nomenclature and units) that explains the typesetting and convention used throughout the paper.

*P4 L23 CHECK EXACT VERSIONING AND TYPESETTING FOR MODEL VERSION OF MITGCM*

We used the MITgcm Release 1 – Checkpoint 65 k at P6L17

*P4 L24 UNCLEAR TYPESETTING, WHY ALL CAPS TRACERS? PLEASE SET PTRACERS IN TYPE- WRITER FONT.*

Done.

*P5 L1FF TYPESETTING. YOU MAY CONSIDER TO INTRODUCE A SPECIFIC TYPESETTING SCHEME FOR YOUR CODE PARTS, BUT YOU NEED TO EXPLAIN IT BEFOREHAND. THIS APPLIES TO THE ENTIRE MANUSCRIPT.*

Thank you for the suggestion. We have introduced a new section 2.1 that explains the convention and typesetting scheme applied to the entire manuscript.

*P5 L1FF CONSIDER TO PUT THIS INFORMATION IN A TABLE.*
Since it is a short list, we would prefer to leave the text as it is.

*P6 L4 WHERE ARE THESE ALTERATIONS "UPGRADES" TO GCHEM DOCUMENTED? WHERE ARE THEY AVAILABLE (AS A PATCH?, IN THE MITGCM DISTRO?)*
These alterations to GCHEM are explained in the BFMCOUPLER manual (appendix A) and the modified routines (GCHEM_CALC_TENDENCY.F, GCHEM_READPARAMS.F, GCHEM_FIELDS_LOADS.F, GCHEM_INIT_FIXED.F and GCHEM_INIT_VARI.F) are provided along with the BFMCOUPLER package in the GIThub repository. In particular, as explained in the manual, a call statement to a BFMCOUPLER routine must be added into the corresponding GCHEM routine.
In the main text we clarified this point by changing the sentence as follows "… by upgrading a few routines of the MITgcm package GCHEM (GeoCHEMistry, details in appendix A)," at P8L10-11.

*P6 L36 THE STAR "*" IS NOT A MATHEMATICAL SYMBOL*
Corrected. Thanks.

*P6 L18 DO NOT EXTEND THE "≈" SIGN OVER THE SUBSCRIPT*
Corrected. Thanks.

*P8 L11,19 INCONSISTENT SUBSCRIPT "S" TO PAR AND MANY MORE EQUATION TYPESETTING PROBLEMS*
Done. We carefully revised the typesetting of all the equations and symbols.

*P9 L35 UNDER WHAT LICENSE IS BFMCOUPLER DISTRIBUTED (MIT)? ALSO NAME THE LICENSE FOR BFM (GPL) AT THE APPROPRIATE PLACE EARLIER IN THE TEXT. HOW DO THESE LICENSES PLAY TOGETHER. DID YOU INCLUDE ANY CODE PARTS FROM BFM INTO THE COUPLER AND HOW DID YOU DEAL WITH CHANGING THE LICENSE?*
The BFM license is a GNU GPL license, and it has been added at P7L30.
Regarding the BFMCOUPLER, we decided for a GPL license in order to be consistent with the license of the BFM model. We believe that our package should be free to be copied, used, modified and published, being consistent with the philosophy of the BFM consortium (bfm-consortium.eu). A license note has been added to the BFMCOUPLER files in the GIThub repository. Thus, the final coupled code will work under the more restrictive license that, to our understanding, is the GPL.

*P9 L35 WHO ENSURES LONG-TERM AVAILABILITY OF THE CODE IF HOSTED ON GITHUB? COULD THE CURRENTLY USED VERSION (WHICH GIT SHA?) BE ARCHIVED AND PROVIDED AS SOM?*
A git SHA-1 hash string, linking the present release 1.0, has been added to the Code availability section in appendix A.
Regarding the long term availability of the code, we would like to point out that other codes published in Geoscientific Model Development are available through GIThub, and we believe that this service will continue ensuring the free availability of our code. However, the long term availability of the code will be also ensured by the modeling group of the OGS Institute (ECHO, http://www.inogs.it/en/content/echo-ecology-and-computational-hydrodynamics-oceanography). Indeed, a copy of the BFMCOUPLER project is now saved in other internal repositories and it is available upon request

*P11 L40 DT_TRC IS NOT EXPLAINED, PLEASE CHECK ALL YOUR ACRONYMS AND SUBSCRIPTS FOR EXISTING EXPLANATIONS. CONSIDER A TABLE OF SYMBOLS.*
The symbol has been corrected, and the sentence revised as follows: "With a $LS_n$ set to 8 (a time step for tracers, $\Delta t_{trc}$, equals 2400 s), the" at P16L7. Further, we would like to remind that we have added a new table (Appendix B) reporting all symbols and variables in order to facilitate the reader.

*P12 L6F PLEASE MOTIVATE YOUR CHOICE OF THE 0–200 M DEPTH RANGE FOR INTEGRATION OF CHL. MAYBE PROVIDE SOME INFOMATION ON THE TYPICAL DEPTH DISTRIBUTION OF CHL IN THIS REGION. ALSO, USE CORRECT TYPOGRAPHY (EN-DASH AND NOT HYPHEN BETWEEN NUMBER AND UNIT).*
The depth of 200 m is the lowest limit of the photic layer in the Mediterranean Sea as reported in Lazzari et al. (2012). Thus, since the idealized case has been designed to reproduce condition similar to those of the Mediterranean Sea, we used this value to compute the integral of the chlorophyll values.

*P12 L22 Provide evidence that the floating point rounding led to this error. How is the mass budget affected by different choices of the LS option?*

We checked this aspect by performing several tests on the time-averaged files dumped by the model I/O routines (output dumped every month, 15 days, 5 days and 1 day) and on the precision of the output files (32 bit versus 64 bit).

The errors on the computation of mass conservation are due to the time discretization/average of the output files (rounding associated with the time average calculations of concentration and SSH). The errors are not due to the floating-point precision used to save the model output, as we erroneously wrote in the previous version of the paper. Therefore, we have redrawn Figure 7 using daily model output and we corrected the sentence as follows: "The errors in mass conservation over time were small ($O(10^{-9})$) and they were caused by the computation of the time average of the model output." at P16L29-30.

*P13 L38 pCO2 not explained, also bad typography.*

pCO2 has been substituted with $pCO_2^{atm}$ in Figure 1 and in the text. The term is explained when introduced at P18L23 and it is listed in the new table of appendix B.

*P14 L5 for how many cores did you come up with a simulation time of 65 hrs per year?*

Thank you for pointing this out. The number of cores was 224, decomposed in a grid of 16 x 14 cores. We have corrected the text.

*P15 L29 I don't agree with the term "framework". You describe a coupling and a coupling strategy, but certainly not a framework.*

The word "framework" has been removed.

*P16 L3ff  This paragraph is entirely unclear to me. Both FABM and MESSy are capa- ble of domain decomposition and have been shown to scale very well. Please better substantiate or argue your reasons not to use these somehow established coupling frameworks.*

We agree: this paragraph was not clear. We simply meant that the efficiency of a coupled code can be an issue. Our coupling (as others like FABM and MESSy) is capable to handle this aspect.

The sentence has been rewritten as follows: "Despite the growth of computational resources, the efficiency of coupled codes can be still an issue because of the large size of the computational grids (Blom and Verwer, 2000). Domain decomposition and parallelization tools are available in several coupling environments (e.g., FABM, Bruggeman and Bolding, 2014; and MESSy, Jöckel et al., 2008). Likewise, our coupling scheme has been thought to fully exploit the parallelization efficiency of the MITgcm (Marshall et al., 1997), and no additional coding effort (in terms of parallelization) is required by the users." at P21L13-17.

*P17 L10 I would recommend to add a section and paragraph "Code availability" here with the web resources, git/archive information, version numbers, and licenses. Move the content from appendix A4 here to this main section.*

Thank you for the suggestion, however, we would rather leave the section A4 in the appendix. The manual (appendix A) and the availability of the code are already mentioned in the main text; and we think that appendix A should end with the indication of the code availability and the description of the experiment that can be downloaded.

However, we will agree to move section A4 into the main text if the reviewer or the editor still recommends doing so.

*Figure 1 Spelling of Omnivorous, Oxidation. From the figure, it is unclear what the superscript numbers (1) etc. denote, please explain. If (as I believe) this is a mere index, then the notation is rather unusual and confusing. Could you find a better notation or make transparent why you use this one? Why do you not use the subscript I for your inorganic nutrients but spell them out, this is inconsistent.*

Thank you for the comment. We made the following changes to the new version of the figure: correction of omnivorous, oxidation; revision of the typesetting of the BFM variables (Helvetica) and subscript for

chemical element (blackboard style), revision of all lines and boxes in order to improve the graphical aspect.

Then, the superscript numbers (e.g. (1)) are mere index used to compose the variable's name in the BFM nomenclature (e.g. $N_P^{(1)}$ equals to `N1p` in the BFM code). This convention is explained in the BFM manual and references thereby. Therefore, in preparing Figure 1, we would prefer to maintain the convention adopted by the BFM consortium since, even if it is not the best notation, it would help the reader to get acquitted with the BFM convention.

Further, the caption has been upgraded as follows: "Figure 1: BFM model: scheme of the functional interactions among the variables in the version that was implemented in Lazzari et al. (2012), Melaku Canu et al. (2015), and Cossarini et al. (2015b). Variable names follow the BFM convention (Vichi et al., 2015). The subscripts indicate the chemical components (C: carbon; P: phosphorus, N: nitrogen, S: silica, O: oxygen)."

*FIGURE 2 THERE IS A NEED TO SYNCHRONIZE ALL SYMBOLS WITH THOSE USED IN THE MAIN TEXT, ALSO REFER TO THE (SUGGESTED) TABLE OF SYMBOLS, IF AVAILABLE. SOME SIMPLIFICATIONS IN TERMINOLOGY COULD HELP, WHY E-P-R, IF EPR IS SUFFICIENT—OR CALL IT MOISTURE BALANCE; WHY THE SUM OF Qx WHERE Qθ IS SUFFICIENT. HELP THE READER DISTINGUISH BETWEEN C FOR CARBON AND C FOR CONCENTRATION, ESPECIALLY AS SUBSCRIPT; MAKE SURE THE NOTATION ALIGNS WITH FIGURE 1.*

The style of the figure has been revised according to the reviewer's suggestions. In the new version of Figure 2, the symbols have been synchronized with those used in the main text, and the typesetting has been resived as well (variables in italic, code and package names in Courier font, name of models and other text in Helvetica font). The sum of heat fluxes has been substituted with the Q; E-P-R has been substituted with *EmPmR* which is the convention used by MITgcm.

A new section 2.1 explains the convention used throughout the text, and a new appendix (Appendix B) reports all the symbols used in the manuscript.

*FIGURE 3 ALIGN TYPESETTING WITH (REVISED) TYPESETTING IN-TEXT. CLEAN UP, E.G., WHY pTRACER AND PTRACER IN THE SAME FIGURE? PLEASE EXPLAIN, HOW YOU NUMERICALLY HANDLE THE EQUATION PTRACER=PTRADERS + DT\*GCHEMTENDENCY WITH REGARD TO CFL AND POSSIBLE UNDER/OVERFLOWS.*

Figure 3 has been revised according to the reviewer's suggestions. In particular, normal text is in Times New Roman, code and routine names are in typewriter style, variables in italic.

All variables (e.g., *pTracer*) have been carefully revised and corrected.

Finally, the numerical integration of the derivative for *pTracer* is handled by the `TIMESTEP_TRACER` MITgcm routine (Euler forward scheme). The CFL condition is intrinsically satisfied in `GAD_CALC_RHS` and in the other routine that calculate the advection and diffusion terms (`GAD_ADVECTION`, or other routine according to the selected option).

*FIGURE 4 THANKS FOR CHOOSING AN ACCESSIBLE COLORMAP! AVOID REDUNDANCY IN GRAPH (OF GEOGRAPHIC LOCATION), DO NOT LET AXIS LABELS OVERLAP. PROVIDE MORE CONTEXT IN THE CAPTION TO WHICH EXPERIMENT (I.E. IDEALIZED CLOSED BASIN) THIS GRAPH BELONGS. MOVE UNIT FROM CAPTION TO COLORBAR; CLARIFY WHETHER THIS IS MAGNITUDE OF 3D SPEED OR MAGNITUDE OF SPEED PROJECTED ON Z-PLANE.*

The figure and the caption have been modified according to reviewer's suggestions.

*FIGURE 5 REMOVE NEGATIVE SIGNS WHEREVER POSSIBLE. DEPTH IS USUALLY A POSITIVE NUMBER INCREASING FROM THE SURFACE DO THE BOTTOM, ADHERE TO THIS MORE COMMON CONVEN- TION. DO NOT DENOTE THE CONTOUR WITH A MINUS SIGN: CONCENTRATIONS AND NPP CANNOT BE NEGATIVE (THAT'S THE FIRST IMPRESSION I GET...). GIVE COLOR-CODED AND CONTOURED QUANTITY AT THE SAME PLACE IN THE FIGURE, LIKE "COLOR: T, CONTOUR: MLD". MAKE SURE YOU EXPLAIN ALL ACRONYMS, SOMETIMES A WORD MIGHT WORK INSTEAD OF THE SYMBOL, E.G. "TEMPERATURE" INSTEAD OF T. BE CONSISTENT WITH UNITS AND EXPONENTS. MOVE UNIT OF COLOR-CODED QUANTITY TO COLORBAR. ROTATE THE "M" UNIT ON Y-AXIS AS IT FALSELY GIVES THE IMPRESSION OF AN "E" AT FIRST GLANCE; ADD "DEPTH" TO Y-AXIS TO HELP THE READER. AVOID OVERLAP OF NUMBERS BETWEEN GRAPHS. SPELL HOVMÖLLER WITH A "V". GIVE CONTEXT ON EXPERIMENT AS IN FIGURE 4. MAKE SURE FONT SIZES ARE SIMILAR TO FIGURE 4, 6 (AND ALL OTHER FIGURES).*

The figure has been redrawn considering all the suggested points. Further, the caption has been rewritten as follows: "Figure 5: Hovmöller diagrams of the (a) Temperature and evolution of the mixed layer depth (MLD), (b) Phosphate and PAR, (c) Chlorophyll (sum of the chlorophyll content in the four phytoplankton functional groups) and Phytoplankton expressed in carbon biomass, (d) Oxygen and Net

Primary Production (NPP), (e) Small Zooplankton (Small Zoopl) and Mesozooplankton (Mesozoopl), and (f) bacteria."

*FIGURE 6 AGAIN, USE CONSISTENT EXPONENTS, EXPLAIN ALL ACRONYMS. ADD MORE MEANING TO THE VERY TECHNICAL CAPTION. PAY ATTENTION TO DETAIL LIKE THE MISMATCH BETWEEN THE STATEMENT "ROOT MEAN SQUARE ERROR" AND RMSD (UNEXPLAINED, BUT I SUPPOSE "DEVIATION"); ADD YOUR CHOICE OF ACCEPTABLE RMSD THRESHOLD TO GRAPH. CONSIDER USING STACKED BARS INSTEAD OF AREA PLOT FOR LEFT AXIS. IMPROVE LOOKS OF FIGURE.*

Figure 6 has been redrawn using stacked bars instead of areas for left axis. Names in the legend are better explained in the caption, as well as RMSD, which is now consistent with the text in the caption. The caption has been rewritten in order to clarify the different terms that appear in the legend.

*FIGURE 7 EXPLAIN "ETA" (OR USE GREEK SYMBOL AS SUGGESTED FOR TEXT, AND ADD SEMANTICS FOR SEA SURFACE HEIGHT), TAUTOLOGY "ALKALINITY CONCENTRATION", USE CONSISTENT NAME "ALKALINITY" OR "TOTAL ALKALINITY". ALIGN STYLE OF FIGURE TO OTHER FIGURES IN YOUR MANUSCRIPT. REPLACE "1 LAYER" WITH "SURFACE LAYER". IMPROVE OVERALL LOOK OF FIGURE (TICKS, LEGEND, LINE WIDTH/STYLES, LABELS, TITLE ,....).*

The figure has been redrawn using daily model output instead of 15-day averages (see comment P12L22) and the general layout has been improved. In particular, Eta has been substituted with Sea Surface Height (SSH), alkalinity label has been corrected, units are consistent with the main text, font size has been increased, and ticks have been aligned.

The caption has been rewritten as follows: Figure 7: Evolution of SSH (blue line) and alkalinity (red line) at the surface layer together with the relative variation of total alkalinity mass (M) with respect to the initial condition (M0) over the whole domain (black line). The total alkalinity mass was obtained by multiplying the daily average model output by the domain volume, which included the time-varying SSH at the surface layer.

*FIGURE 8 IMPROVE OVERALL FIGURE QUALITY. CHOOSE MORE CONTRAST FOR DEPTH COLORMAP (CONSIDER ESTABLISHED TERRAIN/OCEAN FLOOR COLORMAPS); LARGER FONT SIZES THROUGHOUT. UNIT OF GEOGRAPHIC COORDINATES IS DEGREE N OR E. ADD PADDING BETWEEN TEXT AND FIGURE MARGINS. MAKE ALL NAMED FEATURES (RIVER ENTRY POINTS, SITE LOCATION, OTRANTO STRAIT) MORE VISIBLE. POSSIBLY COMBINE WITH FIGURE 9 SHOWING 3 PANELS ALONGSIDE? THE LINE SHOWING OTRANTO STRAIT IS NOT IDENTICAL TO THE EXTENT SHOWN IN FIGURE 11.*

Figure 8 has been completely redrawn. The quality of the figure has been improved. Longitude and latitude labels have been corrected and are now consistent with those of Figure 9. All named features have been made more visible. And the extent of line showing the Otranto Strait has been corrected.

Two new dashed lines have been added to show the two open boundaries: the Sicily Channel and the Cretan Passage.

*FIGURE 9 AVOID REDUNDANCY IN Y-AXIS (LATITUDE); BE CONSISTENT WITH UNIT (SEE FIGURE 8), IT IS NOT HELPFUL TO SHOW 10 TIMES "N" ALONG THE AXIS. S NOT EXPLAINED, AGAIN DIFFERENT NOTATION FOR CHLOROPHYLL (CHLA, IN OTHER PLACES CHL). ADD INFORMATION ON COUNTROU QUANTITY IN THE PLOT (NEAR THE COLORBAR). PANEL A) SURFACE CURRENT MAGNITUDE NEEDS A SCALE BAR. PANEL B) I PREFER CHL TO BE SHOWN ON A LOG SCALE. BLUE CONTOUR UNFORTUNATELY COLLIDES WITH BLUE COLOR IN IONIAN BASIN, TRY DIFFERENT COLOR FOR THIS CONTOUR.*

Figure 9 has been redrawn improving its quality. In particular in plot (b) a logarithmic color scale has been adopted for chlorophyll. This helped to make the blue contour (NPP) more distinguishable from the color plot. Variables are now clearly listed in the plot and their names are consistent with those used in the text.

Further, the caption has been changed accordingly.

*FIGURE 10 NEGATIVE DEPTH AXES, LOG SCALE, COLLIDING LINES AND TEXT, FONT SIZES, MISLEADING Y-AXIS LABEL... PLEASE PAY ATTENTION TO THESE DETAILS. SUGGESTION: ADD "ADRIATIC PIT" AND "IONIAN DEEP" TO A) AND B) TO IMPROVE INTUITIVE ACCESS TO FIGURE. VISUALLY SEPARATE YEARS AND SPELL OUT THE FULL YEAR 2008 ETC. UNDER THE X-AXIS. ADD "CONTOUR" IN CAPTION*

Figure 10 has been changed as follows: convention of depth axes is now consistent with the reviewer suggestion, font sizes have been revised, years have been spelled out and ticks to x-axes added, the labels "Adriatic Pit" and "Ionian Sea" have been added to (a) and (b) respectively.

Then, we tested logarithmic colorscale for chlorophyll, however, the graphical results were not satisfying, therefore we decided to keep the current colorscale.

The caption has been revised as follows: "Hovmöller diagrams of chlorophyll (color) and phosphate (contour, [mmol m$^{-3}$]) and plots of the mixed layer depth (dashed lines, [m]) for the southern Adriatic Pit (a) and the Ionian offshore area (b)."

*FIGURE 11 NEGATIVE DEPTH AXES, COLLIDING LINES AND TEXT AND COLLIDING TEXTS, MISSING AXIS LABELS, UNDEFINED ACRONYMS ... AVOID REDUNDANT X-AXIS (LONGITUDE) INFORMATION, ADD UNITS TO COLORBAR, ADD CONTOUR INFO WITHIN FIGURE. CONSIDER DIFFERENT SCALING OF COLORED QUANTITIES, E.G. LOG OR SQUARE ROOT (WITH WORKAROUND FOR NEGATIVE VALUES) TO SHOW MORE DETAIL IN THE SMALL NUMBERS. ADDED SEMANTICS LIKE "VERTICAL SECTION" TO CAPTION.*

Several changes have been made to Figure 11. In particular, a new colorscale has been adopted. The contour lines of the velocity field are blue and red for southward and northward velocities, respectively. The x-axis of the upper panel has been removed. Legends and texts have been made bigger and more readable.

The caption has been improved as follows: "Fluxes of organic carbon (upper panel) and DIC (lower panel) across the Otranto Strait (dashed line in Fig. 8). The solid contours specify northward (red) and southward (blue) meridional velocities.".

*TECHNICAL CORRECTIONS*

*P2 L7 "IS" SHOULD BE "HAS BECOME"*
Done.

*P2 L40 "FOLLOWING" SHOULD BE "SUBSEQUENT"*
Done.

*P3 L8 DELETE "VERY"*
Done.

*P4 L22 COMPOSED OF*
Done.

*P4 L31F USE A PROPER EM-DASH AND NOT A HYPHEN*
Done.

*P5 L11 "CUSTOM" INSTEAD OF "CUSTOMARY"*
Done.

*P11 L25 WELL-LIT, NOT "WELL-LIGHTED"*
Done.

*P12 L17 "ETA" SHOULD $H$?*
Eta has been substituted with SSH here and throughout the manuscript.

*P13 L35FF USE CONSISTENT EXPONENT, E.G, EITHER M$^{-1}$ OR 1 OR 1/M, BUT ONLY USE ONE OF THOSE STYLES IN THE TEXT. I PREFER THE FIRST WITH NEGATIVE EXPONENT. CHECK ENTIRE MANUSCRIPT.*
The entire manuscript has been checked for consistent exponents (e.g. m$^{-1}$ convention is used)

*P15 L1 "CONSISTENT" NOT "CONSISTENTLY"*
Done.

Reference used in the present letter:

Cossarini, G., Lazzari, P. and Solidoro, C.: Spatiotemporal variability of alkalinity in the Mediterranean Sea. Biogeosciences, 12(6), 1647-1658, 2015.
Melaku Canu D., Ghermandi, A., Nunes, P.A., Lazzari, P., Cossarini, G. and Solidoro, C.: Estimating the value of carbon sequestration ecosystem services in the Mediterranean Sea: An ecological economics approach. Global Environmental Change. 32, 87-95, 2015.

Lazzari, P., Solidoro, C., Ibello, V., Salon, S., Teruzzi, A., Béranger, K., Colella, S. and Crise, A: Seasonal and inter-annual variability of plankton chlorophyll and primary production in the Mediterranean Sea: a modelling approach.Biogeosciences, 9, 217-233, doi:10.5194/bg-9-217-2012, 2012.

Lazzari, P., Solidoro, C., Salon, S. and Bolzon, G.: Spatial variability of phosphate and nitrate in the Mediterranean Sea: A modeling approach. Deep Sea Research Part I: Oceanographic Research Papers, 108, 39-52, 2016.

Sannino, G., Carillo, A., Pisacane, G., Naranjo, C., 2015. On the relevance of tidal forcing in modelling the Mediterranean thermohaline circulation Prog. Oceanogr. 134, pp. 304-329 doi:10.1016/j.pocean.2015.03.002

[revised manuscript text omitted]

---

## Author Comment (AC2) · 15 Feb 2017

We thank the reviewers for their comments and suggestions. Please see the attached document (on Rev#2 response) reporting our responses and a new version of the manuscript with our changes.
* * *